# Bayesian-optimization-assisted discovery of stereoselective aluminum complexes for ring-opening polymerization of racemic lactide

Xiaoqian Wang [1,2], Yang Huang[1,2], Xiaoyu Xie[1,2], Yan Liu[1], Ziyu Huo[1], Maverick Lin[1], Hongliang Xin [1] ✉ & Rong Tong [1] ✉

Stereoselective ring-opening polymerization catalysts are used to produce degradable stereoregular poly(lactic acids) with thermal and mechanical properties that are superior to those of atactic polymers. However, the process of discovering highly stereoselective catalysts is still largely empirical. We aim to develop an integrated computational and experimental framework for efficient, predictive catalyst selection and optimization. As a proof of principle, we have developed a Bayesian optimization workflow on a subset of literature results for stereoselective lactide ring-opening polymerization, and using the algorithm, we identify multiple new Al complexes that catalyze either iso-selective or heteroselective polymerization. In addition, feature attribution analysis uncovers mechanistically meaningful ligand descriptors, such as percent buried volume ($\%V_{bur}$) and the highest occupied molecular orbital energy ($E_{HOMO}$), that can access quantitative and predictive models for catalyst development.

High-performance homogeneous single-site catalysts for polymer synthesis are required for economically producing environmentally friendly degradable polymers[1–4]. Moreover, because the stereochemistry of polymers determines their thermal and mechanical properties, the development of polymerization catalysts that provide stereoregular, microstructure-defined polymers has become increasingly important for synthesizing degradable polymers with new properties and applications[5–13]. For example, poly(lactic acid) (PLA) with stereoregular microstructures, e.g., isotactic structures, is crystalline, and have improved thermal properties (e.g., specific melting temperature) and mechanical properties (e.g., higher tensile modulus) than the atactic PLA[12,14]. However, because of the structural complexity of many metal-based polymerization catalysts, structure–activity relationships are often difficult to interpret[15–18]. Trial-and-error-based discovery and optimization of polymerization catalysts can be both time-consuming and expensive because this method relies on polymer chemists' experience and empirical knowledge, and on serendipity.

The fundamental challenges associated with catalyst discovery are not unique to polymer chemistry, and organic chemists have addressed them by establishing linear relationships between specific reagent descriptors and various outputs (e.g., product selectivity) on the basis of mechanistic hypotheses, such as the Hammett equation that relates chemical structure, originally represented by quantitative experimental parameters, to reactivity[19,20]. Recently, a complementary approach has emerged in the chemistry community that applies data-driven machine learning methods to capture multidimensional structure–activity relationships for catalysts[21–27]. Machine learning approaches can accept numerous reagent features and reaction conditions as inputs without recourse to a specific mechanistic hypothesis, and can recognize hidden patterns in a multidimensional chemical space[26,28]. In particular, the machine learning surrogate

[1]Department of Chemical Engineering, Virginia Polytechnic Institute and State University, 635 Prices Fork Road, Blacksburg, VA 24061, USA. [2]These authors contributed equally: Xiaoqian Wang, Yang Huang, Xiaoyu Xie. ✉e-mail: hxin@vt.edu; rtong@vt.edu

model in Bayesian optimization, e.g., Gaussian process regression (GPR), uses parameter distributions reflecting the uncertainty of physical variables, as opposed to conventional computationally derived point values, and is thereby advantageous for quantifying uncertainty in the exploration process[29,30]. This approach has been successfully used to develop enantioselective catalysts and to predict reaction yields in organic chemistry[28,30–35].

In a typical Bayesian optimization workflow (Fig. 1a), chemical structures are represented by descriptors—mathematical tools for describing properties of subunits or entire molecules—together with parameterized reaction conditions to establish a dataset to train a probabilistic surrogate model, which is constructed by learning from previous observations with a prior over functions[31,36]. After the surrogate model is trained, new experiments are sequentially chosen by optimizing an acquisition function that proposes potentially optimal data points for the next evaluation of the reaction. The chosen experiments are then carried out, and the results are put back into the dataset to update the surrogate model, thereby completing one round optimization[28,32].

However, implementing such an approach in stereoselective polymer chemistry can be particularly difficult because the catalyst dataset for many stereoselective polymerizations is relatively small (usually <100 unique catalysts) compared with the datasets for organic reactions (>1000 unique catalysts)[28,30,33]. In addition, unlike one-step enantioselective organic reactions, stereoselective polymerizations involve hundreds of enantioselective reactions, and the free-energy difference affecting the stereoselectivity can be marginal (in the 2–5 kcal mol$^{-1}$ range)[16,17,37]. Therefore, selecting mechanistically relevant descriptors from multidimensional datasets for machine learning is challenging for polymerization reactions[16,38]. Bayesian optimization has never been used to discover stereoselective polymer catalysts. Moreover, no efficient implementable strategy based on data science has been developed for use as a mechanistic tool for understanding nonintuitive trends in catalyst performance in polymer science.

We hypothesized that the use of Bayesian optimization approach that can efficiently handle small datasets for machine learning might serve as a framework for overcoming the challenges posed in discovering stereoselective polymerization reaction catalysts[39]. Establishing such a framework not only would enable the discovery of new stereoselective polymerization catalysts but also would provide a quantitative tool for rationalizing catalyst performance in mechanistic studies. Herein, we describe a workflow and analysis framework to achieve these goals. We focused on Al-mediated stereoselective ring-opening polymerization (ROP) of racemic lactide (rac-LA), which affords stereoregular PLA (Fig. 1b). Starting from literature data points for tetradentate salen- and salan-type Al complexes, we showed that our Bayesian optimization model can guide the discovery of multiple high-performance isoselective and heteroselective Al complexes for the ROP of rac-LA. Analysis of the machine-learned results revealed important albeit nonintuitive descriptors that can be used for mechanistic studies. Ultimately, our framework serves as an important quantitative tool for both iterative catalyst discovery and mechanism rationalization in polymerization chemistry.

## Results

### Benchmarking the machine-learning algorithms

PLA is an attractive commodity polymer because it is renewable and degradable, and it has numerous applications in packaging, agriculture, and biomedicine[14,40]. Its physicochemical properties are directly related to its tacticity[41,42]. Because producing enantiopure LA monomers carrying two stereogenic centers is difficult, stereoregular PLA is synthesized by ROP of rac-LA using stereoselective metal catalysts[4,43–45]. We focused on symmetrical salen- and salan-type Al

complexes because they are the most frequently studied metal complexes for the ROP of rac-LA[14,41,46], which might provide reasonably sufficient data points to initiate the machine learning process. These Al complexes can also provide stereoregular PLAs with different tacticities, including stereoblock and heterotactic PLAs[45,47]. These complexes usually exhibit isoselectivities in the ROP of rac-LA with decent $P_m$ values ($P_m$, probability of meso linkages)[47,48], and only three salan-Al complexes have been shown to lead to heterotactic PLAs with $P_r$ values exceeding 0.8 ($P_r$, probability of racemic linkages; Supplementary Table 1)[45]. Note that we did not include asymmetrical ligands such as salalen ligands due to computation challenges and possibilities involving complicated stereoselectivity mechanisms (detailed discussion about ligands selection in Supplementary Table 1).

We aimed to use Bayesian optimization to discover new Al complexes with $P_m$ or $P_r$ values exceeding 0.8 ($P_r = 1 − P_m$) for stereoselective ROP of rac-LA. We started by extracting the descriptors from the 56 unique data points for ROP of rac-LA catalyzed by salen- and salan-Al complexes in the literature (Supplementary Table 1)[45,47–56]. Because of the high computational cost of each whole molecule and the large total number of symmetrical ligands (576, Fig. 1a), we utilized a fragmentation strategy[57,58] whereby we divided each catalyst ligand into an arene ring (fragment $A_m$, containing the $R_1$ and $R_2$ groups) and an amine linker (fragment $B_nC_p$, containing the $R_3$ and C groups, Fig. 1a). Results for each fragment were combinatorially concatenated into new vectors to represent the properties of the whole catalyst (Fig. 1a). In addition, using a fragmentation strategy could facilitate structure predictions for late-stage synthesis, whereas using the entire catalyst structure for prediction could initially be much more difficult when having a relatively small dataset.

We applied various methods to generate descriptors for the machine learning surrogate model, and we benchmarked these descriptors' performance on these 56 data points from literature. These descriptors include one-hot-encoding[59], electrotopological-state index[60], eigenvalues of coulomb matrix[61], molecular characteristics generated by the Mordred program[62], and properties obtained by DFT calculations using the Gaussian program[63] at the B3LYP-D3/6-31G(d)/SMD (toluene) level of theory (details in Supplementary Information S4.1–4.2)[64–67]. We built upon the algorithms developed by Doyle and coworkers[30] (auto-QChem in Github) to extract DFT descriptors for machine learning. These descriptors were then evaluated on regression performance using the GPR surrogate model[68]. GPR is a popular probabilistic machine-learning regression model for Bayesian optimization. For continuous domains such as stereoselectivity, it is typical to assume that the unknown function can be sampled by means of a Gaussian process (details in Supplementary Information S4.3)[68]. For each type of descriptors, we applied 5-fold cross validation for the training, that is, in each fold, we divided the randomly shuffled dataset into 45 data points as training set and 11 points as test set. We found that consistent regression performance—lowest mean errors and standard deviations—could be achieved using the datasets generated by electrotopological-state index, Mordred and DFT (Fig. 2a; details in Supplementary Information S4.5). Notably, DFT-encoded descriptors, whose parity plot between GPR-predicted and measure $P_m$ values shown in Fig. 2b, could provide rich chemical information with insights for reaction mechanism studies. Therefore, we carried out the remainder of the studies using DFT-encoded descriptors.

Next we investigated the searching efficiency of Bayesian optimization, implemented using the algorithms by Doyle and coworkers[30] (edbo in Github), and benchmarked its searching performance against a random search process still using the 56 data points from the literature. We carried out the optimization to search the points with the highest $P_m$ or $P_r$ values in the literature dataset over 12 iterations (defined as one run). For both methods, 3 initial points were randomly selected, and 3 new points were proposed per iteration (details in Supplementary Information S4.5). The whole optimization process was independently repeated for 10 runs. For both $P_m$ and $P_r$

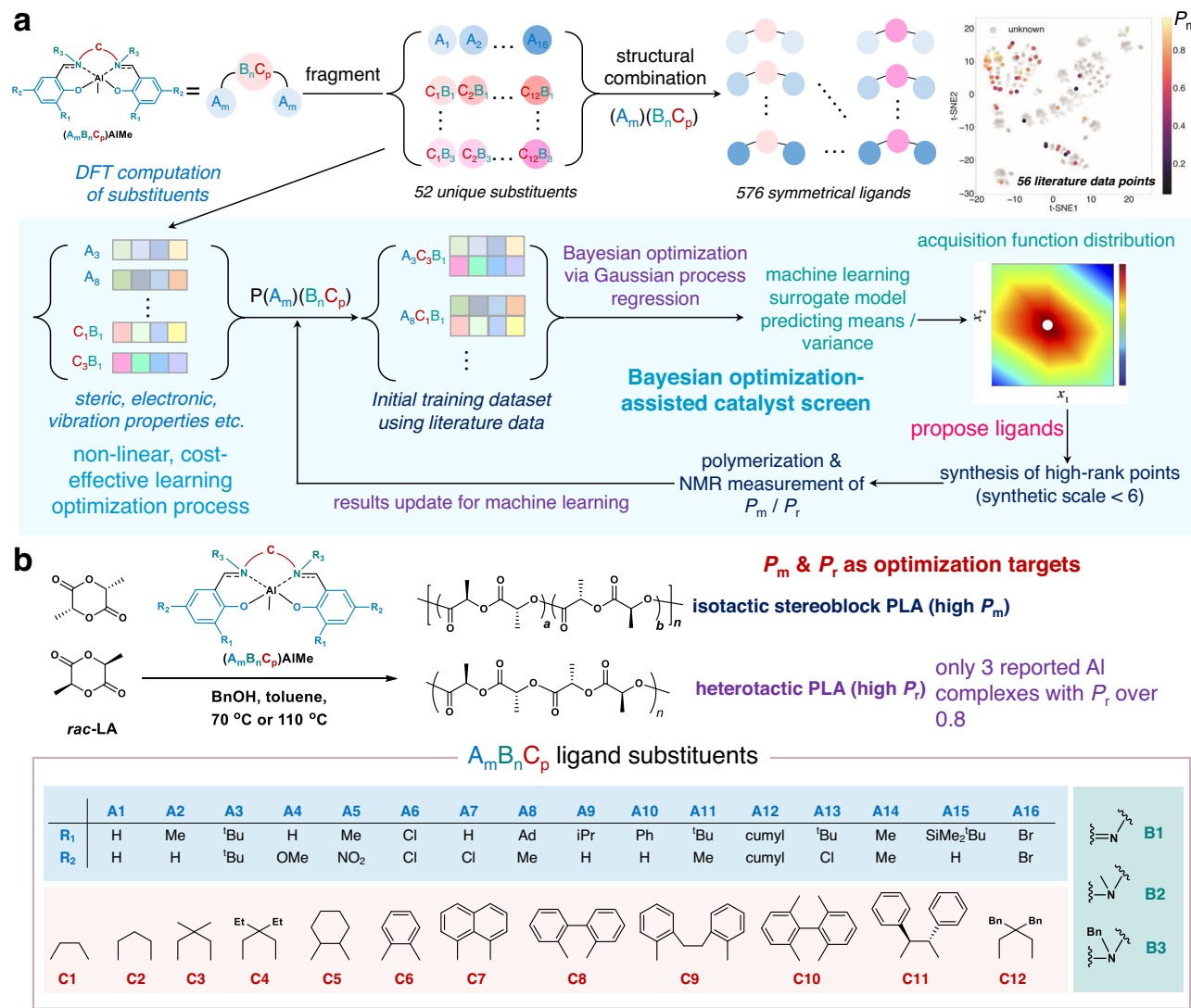

**Fig. 1 | Summary of Bayesian-optimization-guided workflow for discovering stereoselective polymerization catalysts. a** Overview of Bayesian optimization. To efficiently utilize computational resources, the $A_m$ and $B_nC_p$ fragments of salen and salan ligands for Al complexes were subjected to DFT calculations to generate descriptors, and were concatenated to generate the datasets of symmetrical ligands. An initial 56-data-point literature dataset, whose distribution over the entire chemical spaces was plotted (details in Supplementary Fig. 5), was used to train a Gaussian process regression (GPR) surrogate model to generate an acquisition function that proposes new experiments in the design space. The experimental results were fed back into the dataset to iteratively optimize the acquisition function and refine the model. **b** Substituents used for salen and salan ligands for stereoselective ring-opening polymerization of racemic lactide.

optimizations, Bayesian optimization converged (i.e. standard deviation of reached zero) within 7 iterations; whereas no convergence was achieved within 12 iterations for the random search process (Fig. 2c, d). Thus, our Bayesian optimization model offered superior search efficiency over the random search process in our case. Additionally, our Bayesian optimization method was also found more efficient to reach convergence, compared to the sequential model-based algorithm configuration method[69] (see Supplementary Information S4.9).

**Algorithm-guided search of stereoselective ROP catalysts**
We then investigated the ability of our model, which was initially trained on the 56 literature data points, to discover new Al complexes with high $P_m$ or $P_r$ values (Fig. 1b). The localized initial data distribution over the entire chemical spaces (Supplementary Fig. 5; details in Supplementary Information S4.6) indicated that multiple iterations were needed in order to approach global optima. Two Bayesian optimization models were built for $P_m$ and $P_r$, respectively, owing to the distinctive two Bayesian optimization directions. We used the trained surrogate model to optimize the expected improvement acquisition

function that balances exploration and exploitation in the discovery process and is built with both the GPR-predicted mean and variance values (details in Supplementary Information S4.4). The model subsequently proposed ligands potentially having high $P_m$ or $P_r$ values. The predicted points were ranked by the acquisition function, and we selected the top-ranked data points (i.e. the most promising ligands proposed by the model) to prepare ligands and verify their stereoselectivities in ROP of *rac*-LA. To circumvent prohibitively multistep, reagent- and time-consuming syntheses, we assigned each substituent a metric called "synthetic scale", which was based on the sum of the expected number of steps required to synthesize $A_m$ and $B_nC_p$ and thus to build the whole Al complex (Supplementary Table 2). We focused on ligands that could be prepared in no more than three steps because we prioritized accelerated catalyst discovery over exploration of whole chemical spaces, and time presents a substantial cost (note that some synthetic-demanding ligands, e.g., $A_{15}C_3B_1$ requiring 5 steps to prepare–even its Al complex having an excellent isoselectivity[48], would not be considered owing to time and materials limitations). The experimental data points obtained for the proposed ligands in each

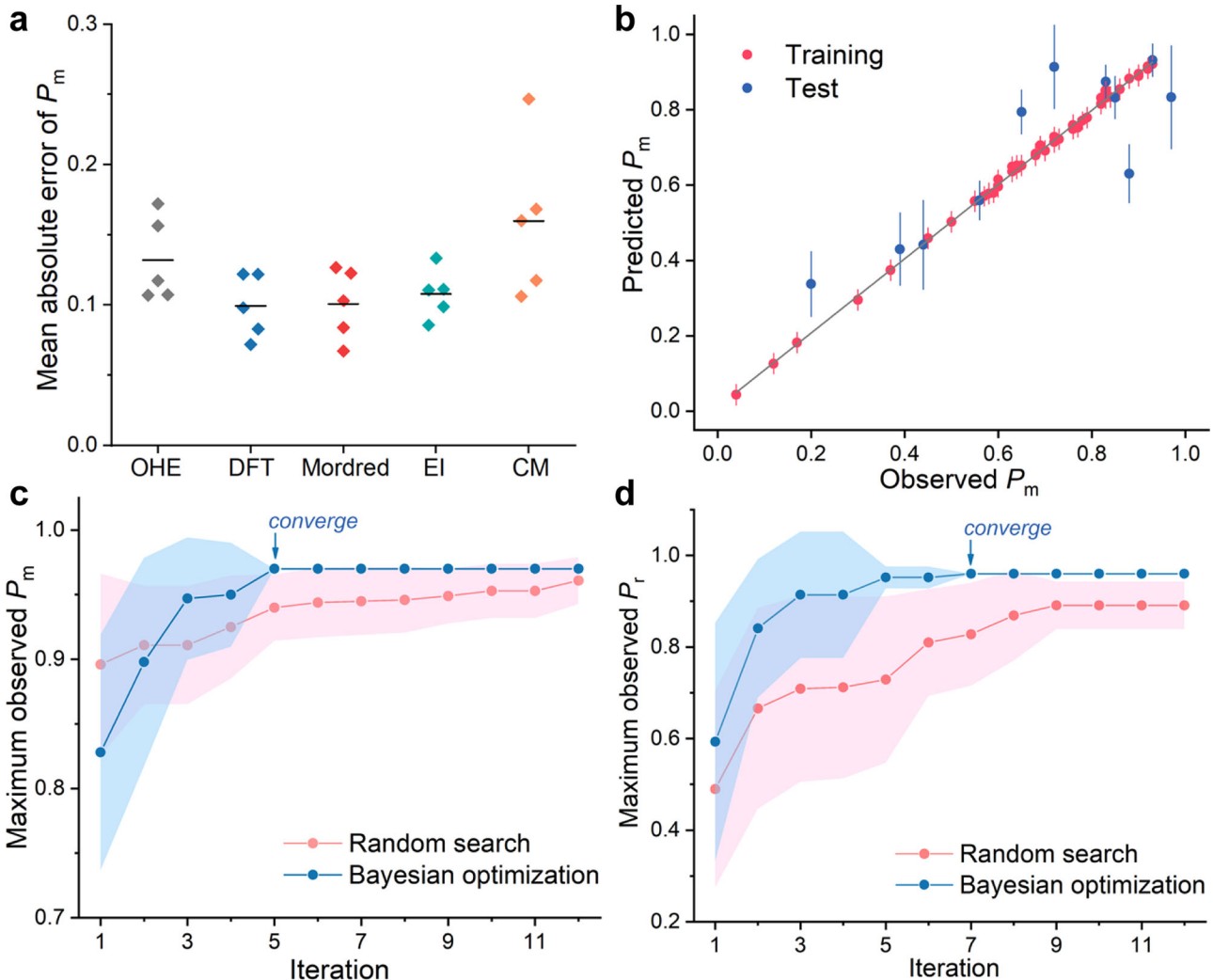

**Fig. 2 | Benchmarking the machine learning algorithms. a** Comparison of machine-learning regression performance using descriptors generated by different methods. The error bars are the standard deviations of prediction errors in the 5-fold cross validations. OHE one-hot encoding, DFT density functional theory, EI electrotopological-state index, CM coulomb matrix. **b** Parity plot of $P_m$ values ($P_m$, probability of *meso* linkages) predicted by Gaussian process regression (GPR) using the DFT-encoded descriptors and observed $P_m$ values obtained from the literature dataset. The error bars are the predicted standard deviation values. The optimization curves for 12-round search of the maximum observed (**c**) $P_m$ and (**d**) $P_r$ values ($P_r$, probability of *racemic* linkages). Each optimization process was

independently repeated for 10 runs (12 iterations per run). For each run, three initial points were randomly selected, and three new points were proposed per iteration. Data are shown as the mean value with the standard deviation (band width) of the highest observed (**c**) $P_m$ or (**d**) $P_r$ up to each iteration (details in Supplementary Information S4.5). The Bayesian optimization curves in **c** and **d** both achieved convergence (i.e. the blue band diminished, pointed with the arrows) within 7 rounds. In contrast, the random search process exhibited large standard deviations (i.e. the red band never diminished) and failed to converge within 12-round optimization.

round were appended to the dataset to refine the model for next-round prediction (Fig. 1a).

We prepared 33 salen- and salan-Al complexes that were proposed by the model. In our first-round modeling, we included polymerization reaction parameters such as temperature and monomer concentration, but these parameters were found to be less relevant to the results of predicted points compared with the catalyst descriptors. This is because the chemical structures of the ligands largely determined the stereoselectivity in polymerization. Therefore, DFT-based descriptors were exclusively used over three computation–experiment rounds, and we carried out the ROP of *rac*-LA in toluene at 70 or 110 °C. Among the 33 newly synthesized complexes, we identified 8 that were iso-selective ($P_m > 0.8$) and 5 that were heteroselective ($P_r > 0.8$; Fig. 3a, b and Supplementary Tables 3–8). The Al complexes with $A_{11}C_3B_1$ and $A_{11}C_2B_1$ both afforded stereoblock copolymers with $P_m$ values over 0.92 (representative homodecoupled $^1$H NMR of the α-methine region in PLA prepared using ($A_{11}C_3B_1$)Al complex in Fig. 3c; NMR spectra of

the PLA synthesized by other Al complexes in Supplementary Figs. 6–11), and high monomer conversions >95% over 12 h ([*rac*-LA]/ [Al] = 100/1, Table 1, entries 1–2), which exhibited slightly better iso-selectivity control compared with the previously reported ($A_3C_3B_1$)Al complex[48] (Fig. 3c and Table 1, entries 1 versus 7).

Additionally, our algorithm-guided approach helped us to discover multiple highly heteroselective Al catalyst ligands, including $A_5C_1B_2$, which had a $P_r$ of 0.94 (homodecoupled $^1$H NMR in Fig. 3c), and $A_5C_1B_3$, $A_{16}C_1B_2$, and $A_{16}C_1B_3$, all of which had $P_r$ values of 0.93 (Fig. 3b and Table 1, entries 3–6). Among the complexes that produced stereoregular PLAs, Al complexes with $A_5C_1B_2$ and $A_{16}C_1B_2$ afforded polymers that exhibited molecular weights close to the calculated values (Table 1, entries 3 and 5) and that showed narrow molecular weight distributions ($Đ < 1.1$; representative size exclusion chromatography in Supplementary Fig. 12), features that are characteristic of well-controlled living polymerization. All of these four Al catalysts showed markedly improved stereocontrol in the preparation of highly

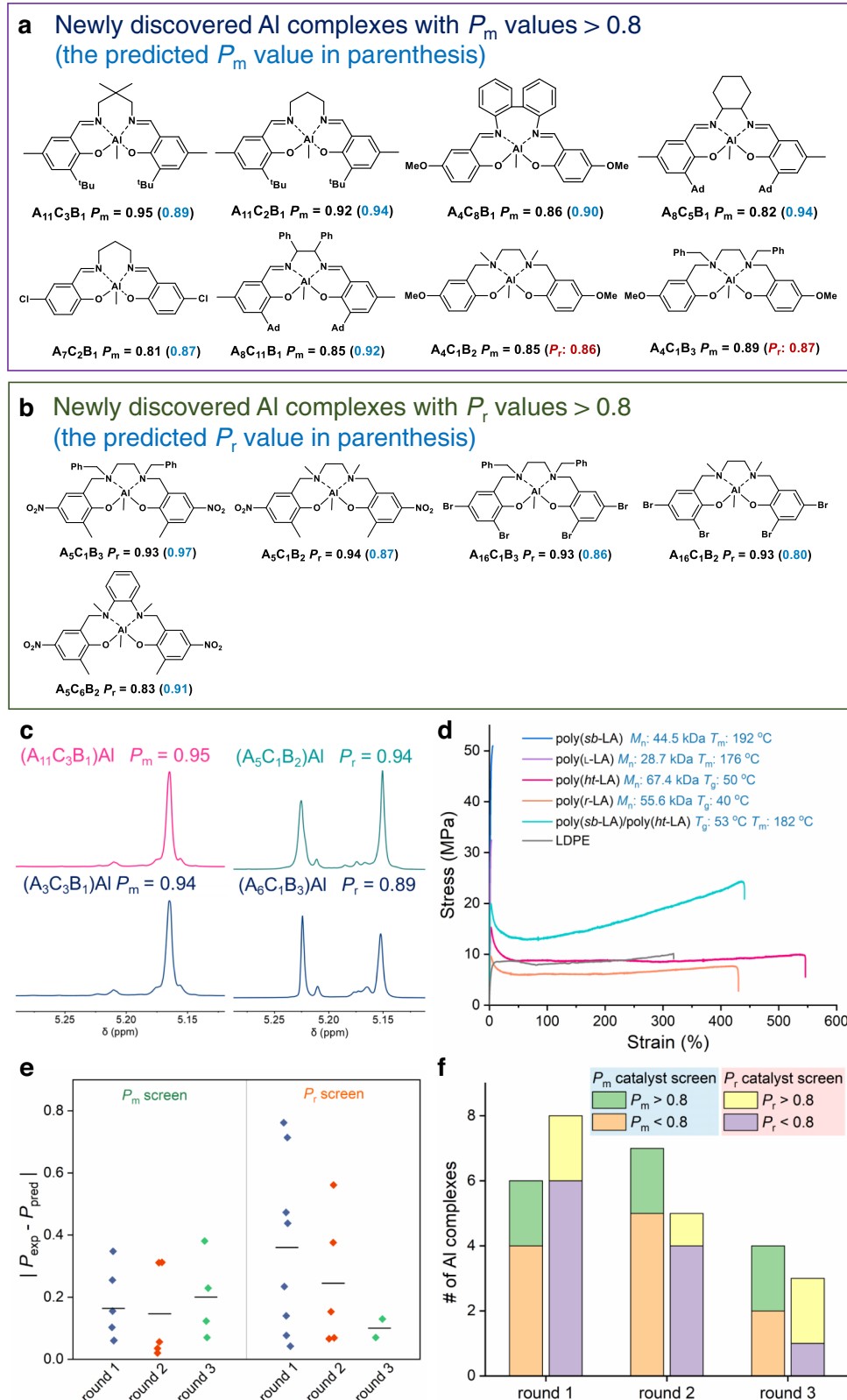

**Fig. 3 | Discovery of stereoselective Al complexes guided by a Bayesian optimization model. a** Al complexes with high $P_m$ values (>0.8; $P_m$, probability of *meso* linkages). **b** Al complexes with high $P_r$ values (>0.8; $P_r$, probability of *racemic* linkages). **c** Representative of homodecoupled $^1H$ NMR spectra of α-methine region in poly(lactic acids) (PLAs) prepared by isoselective catalyst $(A_{11}C_3B_1)Al$ (left), and heteroselective catalyst $(A_5C_1B_2)Al$ (right), and both are compared to PLA prepared by reported catalysts $(A_3C_3B_1)Al$ and $(A_6C_1B_3)Al$. **d** Representative

stress–strain curves obtained by uniaxial extension of PLAs with various micro-structures, and low-density polyethylene (LDPE). Polymer molecular weights, molecular weight distributions, and phase-transition temperatures are provided in Supplementary Table 9. **e** Absolute error between predicted and experimental values ($|P_{exp} - P_{pred}|$) of catalysts identified in each round of optimization. **f** The number of highly stereoselective complexes discovered in each round of optimization.

**Table 1 | Representative stereoselective Al complexes for ring-opening polymerization of racemic lactide (*rac*-LA)**

| Entry | Ligand | Predicted mean $P_m$ | Measured $P_m$ ($P_r$)[a] | Conv %[b] | $M_n$ (kDa)[c] | $MW_{cal}$ (kDa) | $Đ$[c] |
|---|---|---|---|---|---|---|---|
| 1 | $A_{11}C_3B_1$ | 0.89 | 0.95 (0.05) | 95.2 | 14.0 | 13.8 | 1.03 |
| 2 | $A_{11}C_2B_1$ | 0.94 | 0.92 (0.08) | 97.0 | 32.2 | 14.0 | 1.05 |
| 3 | $A_5C_1B_2$ | 0.13 | 0.06 (0.94) | 78.1 | 11.1 | 11.3 | 1.01 |
| 4 | $A_5C_1B_3$ | 0.03 | 0.07 (0.93) | 81.3 | 8.3 | 11.7 | 1.06 |
| 5 | $A_{16}C_1B_2$ | 0.20 | 0.07 (0.93) | 87.7 | 12.9 | 12.6 | 1.04 |
| 6 | $A_{16}C_1B_3$ | 0.14 | 0.07 (0.93) | 58.8 | 15.9 | 8.5 | 1.08 |
| 7[d] | $A_3C_3B_1$ | – | 0.94 (0.06) | 96.2 | 11.9 | 13.9 | 1.16 |
| 8[d] | $A_6C_1B_3$ | – | 0.11 (0.89) | 75.8 | 12.5 | 11.0 | 1.09 |

*Conv* conversion, $M_n$ number-average molecular weight, $MW_{cal}$ molecular weight calculated from feed ratio and LA conversion, $Đ$ molecular weight distribution, $P_m$ probability of *meso* linkages, $P_r$ probability of *racemic* linkages. Polymerization conditions: [*rac*-LA]/[Al] = 100/1 at 70 °C for 12 h; [*rac*-LA] = 1.39 M in toluene.
[a]Determined by [1]H NMR and [13]C NMR spectroscopy. $P_r$ = 1−$P_m$. See Supplementary Information S1.2 for details.
[b]Determined by [1]H NMR spectroscopy.
[c]Determined by size exclusion chromatography (SEC).
[d]Al complexes reported in the literature[45,48] were prepared for comparison.

heterotactic PLA, when compared to the previously reported ($A_6C_1B_3$) Al complex[45] (Table 1, entries 3–6 versus 8).

Furthermore, differential scanning calorimetry measurements of a stereoblock PLA with a $P_m$ of 0.96 (prepared using ($A_{11}C_3B_1$)AlMe complex; $M_n$ = 44.5 kDa, *sb*-PLA) had a melting temperature ($T_m$) of 192 °C; whereas a heterotactic PLA with a $P_r$ of 0.87 (prepared using ($A_{16}C_1B_3$)AlMe complex; $M_n$ = 67.4 kDa, *ht*-PLA) exhibited a glass transition temperature of 50 °C (Supplementary Table 9; representative differential scanning calorimetry results in Supplementary Fig. 13), values that are consistent with the literature[56,70]. We also characterized the stress–strain characteristics of the synthesized stereoregular PLAs. The *sb*-PLA exhibited a fracture strength (σ) of 48.5 MPa and a fracture strain (ε) of 5.0%; whereas the *ht*-PLA—whose physico-mechanical property has been underexplored—exhibited the elastomeric behavior with a high ε of 533% and a σ of 11.2 MPa (Fig. 3d). Additionally, we found the blend of *sb*-PLA and *ht*-PLA at the 1/1 mass ratio showed improved ductility and toughness compared to *sb*-PLA, and outperformed the non-degradable commodity low-density polyethylene (Fig. 3d). These results suggest that the microstructures of polymers impact the thermo-mechanical properties, and mixing PLAs with different tacticities may improve their mechanical properties[71,72].

More important, over the course of algorithm-guided discovery, the mean absolute error between the experimental and predicted values dropped markedly, from 0.36 in the first round to 0.10 in the third round in the search for complexes with high $P_r$ values (Fig. 3e). The portion of high-performance stereoselective Al catalysts ($P_m$ or $P_r$ > 0.8) discovered in the third iteration is higher than the first and second iterations, suggesting an improved search efficiency of our model (Fig. 3f). The less accuracy of the first-round $P_r$ prediction can be observed in $A_4C_1B_2$ and $A_4C_1B_3$, which were predicted as heteroselective catalysts but turned out to be isoselective ones. We reason that the limited heteroselective catalyst information—only 9 of 56 complexes having $P_r$ values over 0.5—at the beginning of the search could contribute to the high prediction error in the first round. Nevertheless, such high uncertainty in the unexplored high-$P_r$-value region in the chemical space quickly decreased when several high-$P_r$-value complexes were identified and appended into the training set to refine the model, thereby significantly decreasing the prediction errors in the subsequent rounds. In particular, the two heteroselective ligands based on $A_5$, which had only one $P_m$ data point ($P_m$ = 0.76) in the literature[54], would not likely have been investigated on a trial-and-error, screening, or intuition-guided study. Despite the small size of the initial training dataset, the performance of our current algorithm

demonstrates that our integrative framework is efficient and capable of proposing valuable data points.

## Attribution analysis and mechanistic studies

The purpose of using data science techniques is not only to discover new catalysts but also to improve our understanding of stereoselective polymerization in such a way as to facilitate catalyst design. Therefore, we utilized the SHAP (SHapley Additive exPlanations) package[73,74] (https://github.com/slundberg/shap), a game theory approach, to analyze the magnitude of each DFT descriptor's contribution to the $P_m$ or $P_r$ value (details in Supplementary Information S4.7). The SHAP value generated in SHAP analysis, which is referenced to the output average values, measures the importance of the individual feature in a coalition of features that cooperate towards forming a prediction in the machine learning model (Fig. 4a)[22,75]. The positive and negative SHAP value refers to positive and negative correlation, respectively, between $P_m$ and the corresponding descriptor. The larger the absolute value, the stronger the correlation is. Notably, the SHAP analysis based on our GPR surrogate model allowed us to identify more descriptors contributing to the stereoselectivity, compared to the analysis using the random forest regression[76] surrogate model (details see Supplementary Information S4.9). We found that a high SHAP value of the minimum %$V_{Bur}$ (percent buried volume[77]) of the $B_nC_p$ fragment contributed the most to the increased $P_r$ values; whereas the maximum %$V_{Bur}$ of the $A_m$ fragment correlated with increased $P_m$, suggesting that the steric effects of the various fragments affected the overall stereoselectivity. Indeed, global analysis of %$V_{Bur}$ for whole Al complexes revealed that highly heteroselective salen and salan Al complexes ($P_r$ > 0.8) had %$V_{Bur}$ values in a narrow range around 67–68%; whereas the %$V_{Bur}$ values for highly isoselective salen and salan Al complexes ($P_m$ > 0.8) were in a relatively broad range of 60–66% (Fig. 4b and Supplementary Table 10). Additionally, SHAP analysis showed that the mean frequency, the Mulliken charge, and the energy of the highest occupied molecular orbital (HOMO) of the $A_m$ fragment contributed to the outcome $P_m$ values in the model (Fig. 4a). To delineate the contributions of the molecular descriptors to the catalyst selectivity, we performed multivariate linear analysis to convert the most important steric and electronic factors into readily interpretable factors (Fig. 4c; details in Supplementary Information S4.8). This analysis revealed decent correlations between the experimentally observed $P_m$ outcomes and the multivariate model containing six electronic descriptors and two steric descriptors from the $A_m$ and $B_nC_p$ fragments ($R^2$ = 0.65). In addition to the steric factors, our attribution analysis showed that altering the electronic properties, especially in the $A_m$ fragment in the Al complexes, could result in significant changes in stereoselectivity. Indeed, the electron-withdrawing substituents in the $A_m$ fragment, which lowered the HOMO energy and increased the electronegativity of the fragment, likely contributed to the increased $P_r$ value (Supplementary Fig. 14a). Multivariate regression analysis of salan-type Al complexes having $A_mB_2C_1$ and $A_mB_3C_1$ ligands (14 ligands), which exhibit a broad range of $P_r$ value, quantitatively described the electronic perturbation on the catalyst stereoselectivity ($R^2$ = 0.94, Fig. 4d), in addition to the steric effect (%$V_{bur}$). Moreover, we evaluated the electronic effects of $B_nC_p$ fragments using Al complexes with $A_3B_nC_p$ ligands (13 ligands), and the multivariate regression model effectively describing that a decrease in the HOMO energy of the $B_nC_p$ fragment—presumably caused by delocalized electrons—contributed to increased $P_m$ values ($R^2$ = 0.89, Fig. 4e; also see model in Supplementary Fig. 14b only including %$V_{bur}$ and $E_{HOMO}$).

In contrast, studying the transition state energies determined on the basis of the established DFT-computed mechanism for stereoselective ROP of *rac*-LA[37] was expensive at high computational cost. The stereoselective ROP of LA proceeds via two transition states: TS1

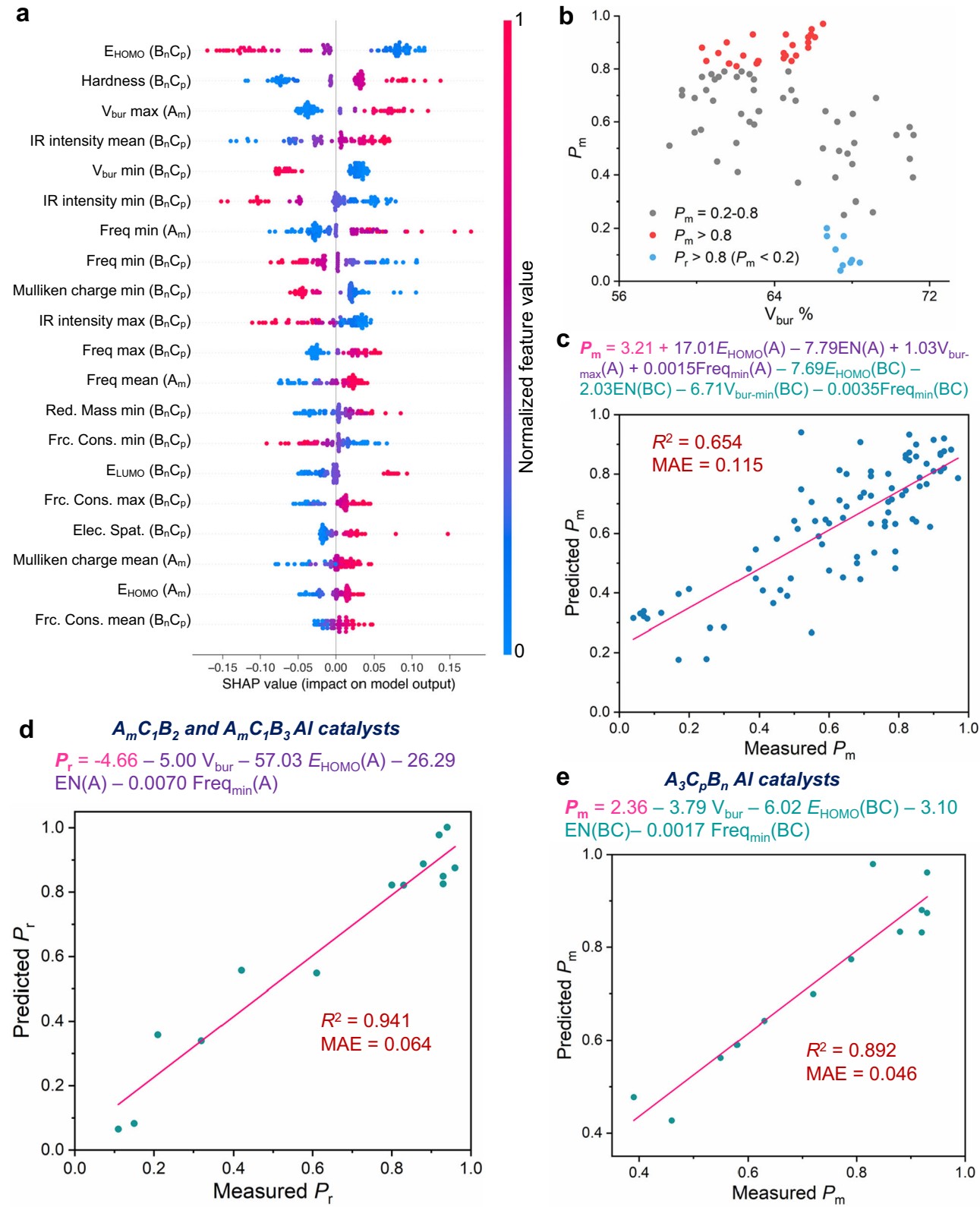

involving nucleophilic attach of Al-alkoxide at carbonyl on the coordinated LA, and a concerted cyclo-reversion reaction in TS2 (Supplementary Fig. 15)[37]. The computation of TS1 states for various Al complexes (at B3LYP-D3/6-31G(d)/SMD (toluene) level of theory) showed that lactide "docking" on the top of Al complex was found energetically favorable in reactions mediated by heteroselective Al complexes, whereas the oligo(LA) resided on the top of Al complex in

reactions involving isoselective Al complexes (Supplementary Fig. 16). This observation confirming the structural importance of %$V_{bur}$, which was highlighted by our SHAP analysis, impacting the stereoselectivity (Fig. 4a, b). Furthermore, our DFT computation results agreed well with previous postulation that TS1 is more relevant transition state for isoselective ROP of LA[37]: the free energy differences between the TS1 for D-LA/ligand-Al-(L-LA) and L-LA/ligand-Al-(L-LA) reactions and the

**Fig. 4 | Attribution analysis of the Bayesian optimization model. a** Descriptor ranking by SHAP (SHapley Additive exPlanations) values based on the Gaussian process regression (GPR) model for optimizing $P_m$ values ($P_m$, probability of *meso* linkages) of the entire dataset of Al complexes. The top rank indicates the most significant effects across all the predictions. The positive and negative SHAP value refers to positive and negative correlation, respectively, between the measured $P_m$ value and the corresponding feature. The larger the absolute value, the stronger the correlation is. The color coding indicates normalized high (red) to low (blue) feature values. **b** Correlation between $P_m$ and %$V_{bur}$ of the Al complex with the $P_m$ value. %$V_{bur}$, percent of buried volume. **c**–**e** Multivariant regression model highlighting important descriptors impacting **c** the $P_m$ values for all Al complexes, **d** the

$P_r$ values (probability of *racemic* dyad formation) in Al complexes with $A_mC_1B_2$ and $A_mC_1B_3$ ligands, and **e** the $P_m$ values in Al complexes with $A_3C_pB_n$ ligands. $E_{HOMO}(A)$, HOMO energy of the $A_m$ fragment (HOMO, the highest occupied molecular orbital); EN(A), electronegativity of the $A_m$ fragment; $V_{bur}$, %$V_{bur}$ of the whole ligand; $V_{bur-max}(A)$, maximum %$V_{bur}$ of the $A_m$ fragment; $Freq_{min}(A)$, minimum frequency of the $A_m$ fragment; $E_{HOMO}(BC)$, HOMO energy of the $B_nC_p$ fragment; EN(BC), electronegativity of the $B_nC_p$ fragment; $V_{bur-min}(BC)$, minimum %$V_{bur}$ of the $B_nC_p$ fragment; $Freq_{min}(BC)$, minimum frequency of the $B_nC_p$ fragment; MAE mean absolute error. In the equations in **c**–**e**, the descriptors of the $A_m$ fragment are highlighted in purple, and those of the $B_nC_p$ fragment in green, and residue numbers in pink.

observed $P_m$ values for highly isoselective Al complexes were usually higher than ~3 kcal mol$^{-1}$ (Supplementary Table 11). Additionally, the relatively large energy barrier differences in the ring-opening of D- and L-LA in TS2 were found in heteroselective Al complexes (Supplementary Table 12), confirming that the heteroselectivity likely is determined by TS2. Nevertheless, such DFT computation of transition states was costly, and was provided as an afterthought for the reaction development, especially in this case where multiple conformers needed to be examined. Together, our work highlights the ability of machine learning to readily identify important mechanistic descriptors for the catalyst search—such as %$V_{bur}$ and $E_{HOMO}$—that also help quantitatively depict complex structure–reactivity relationships, without requiring extensive DFT computation resources and the knowledge of transition states.

## Discussion

Rationally optimizing catalysts is often challenging. Moreover, once a catalyst works, it may not be clear which specific features of the catalyst underpin the performance. Herein we present a holistic, data-driven workflow that can be used to discover high-performance catalysts for stereoselective ROP and to understand the structural factors that impact catalyst stereoselectivity. Our findings clearly demonstrate the power of machine learning techniques for accelerating catalyst development with proposals that might be outside scientists' intuition while also minimizing time and material costs. The workflow's capability of efficiently proposing high-performing stereoselective polymerization catalysts has the potential to change the way polymer chemists select and optimize catalysts. Information about the mechanism of a given polymerization is rarely fully accessible in the early stages of the catalyst search; therefore, our workflow would be especially effective when data for iterative ligand searches is relatively sparse, especially when commercial ligands provide only modest performance at the beginning of a study. Additionally, our workflow can also be used to quantify the contributions of various determinants of catalyst performance. We expect that in the future, deep learning models (e.g., graph neural networks[78]) could be incorporated into our workflow to enhance search efficiency for more flexible catalyst scaffolds, in the discovery of chiral catalysts for enantioselective polymerization of racemic monomers[79–83], and in the synthesis of stereosequence-defined polymers from mixtures of monomers[8].

## Methods
### Polymerization procedures

In a glove box, *rac*-lactide (200 mg, 1.39 mmol) in the toluene (1.0 mL) was mixed with benzyl alcohol (1.5 mg, 0.014 mmol) and Al catalyst (0.014 mmol) in a 15 mL thick-wall glass vessel equipped with a stirrer bar ([*rac*-LA]/[Al]/[BnOH] = 100/1/1). The reaction was stirred at 70 °C for overnight. The reaction was cooled to room temperature, and an aliquot of the solution was dried for NMR analysis to determine the conversion and stereochemistry (see Supplementary Information S1.2). The remaining solution was dried, and the obtained solid was washed by excess methanol to remove the residue monomers for SEC analysis (see Supplementary Information S1.3).

### DFT descriptor generation for machine learning

Descriptors for each of the two fragments of each whole molecule are calculated before they are concatenated to form a single fixed-length feature vector to represent the whole molecule. We modified *autoqchem* package[30] (https://github.com/PrincetonUniversity/auto-qchem) to generate below descriptors from Gaussian output files. The DFT descriptors include: the number of atoms, charge, spin multiplicity, dipole moment, electronic spatial extent, self-consistent field energy, values and corrections of *E, H, G*, ZPE, stoichiometry, HOMO, LUMO, electronegativity, hardness, element labels, atomic buried volume, atomic Mulliken charge, atomic polar tensor charge, vibrational frequencies, reduced mass, force constants, IR intensity and steric descriptors. Steric descriptors (e.g., sterimol[84]) for each molecule were measured by MORFEUS (github.com/kjelljorner/morfeus) using on the Gaussian 16 output results. Source code, DFT descriptor sets, and descriptor computation scripts can be found in https://github.com/hlxin (https://doi.org/10.5281/zenodo.7982855).

### Bayesian optimization

The initial dataset that trained the machine learning GPR surrogate model were 56 data points (i.e. 56 unique Al complexes) collected from literature (see Supplementary Table 1)[45,47–50,52–55]. As shown in Fig. 1, the Al complexes were fragmented into substituents $A_m$ and $B_nC_p$. Descriptors corresponding to each substituent in these 56 data points were generated by various methods (see Supplementary Information S4.2), and combined together for Bayesian optimization using GPR model using re-coded *edbo* package (source code see https://github.com/hlxin). Before being fed into the machine learning model, features were normalized and decorrelated to reduce the chance of overfitting, by removing features with Pearson correlation coefficient over 0.95 (a discussion of feature interdependency is provided in Supplementary Information S4.10). Expected Improvement acquisition function is used for new Al-complex molecules selection in each iteration (more details in Supplementary Information S4.5). Considering the complexity of new molecule synthesis, a metric called "synthetic scale" is proposed, based on the sum of evaluated number of synthesis steps of each of the two split small molecules of each whole Al-complex. Values from low to high correspond to the increase in synthesis difficulty, and we only synthesize the model-suggested catalysts with low synthetic scale values (see Supplementary Table 2). This can be regarded as an extra component of the acquisition function in addition to mean and variance. For the three iterations of Bayesian optimization, all the data of model-proposed catalysts is tabulated in Supplementary Tables 3, 5, and 7 for isoselective catalysts discovery and Supplementary Tables 4, 6, and 8 for heteoselective catalysts discovery.

### SHAP analysis

We used the SHAP (SHapley Additive exPlanations) package (https://github.com/slundberg/shap) a game theoretic approach to explain the output of a machine learning model, to calculate the Shapley value, the magnitude of contribution of each normalized feature in DFT representation for the determination of $P_m$ or $P_r$ referenced to output

averages. The principle of SHAP analysis was described in Supplementary Information S4.7.

## Multivariate linear regression

Multivariate linear regression analysis was performed using R 4.1.2. The descriptors for the model were selected based on the SHAP analysis, selecting most impactful descriptors that were mechanistically meaningful for analysis.

## Reporting summary

Further information on research design is available in the Nature Portfolio Reporting Summary linked to this article.

## Data availability

Source data, including dataset of DFT coordinates as Supplementary Data 1, are provided with this paper. The authors declare that the data supporting the findings of this study are available within the Article and its Supplementary Information file, or from the corresponding author upon reasonable request. Source data are provided with this paper.

## Code availability

The source code for descriptor generation, the computation scripts for machine learning, all of the DFT descriptor sets, and other descriptor sets, including one-hot-encoding, Mordred, CM and EI descriptors, and Gaussian output files used in this work can be found in https://github.com/hlxin/bayespoly, or in Zenodo[85].

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

## Acknowledgements

This work is made possible by the use of Virginia Tech's Materials Characterization Facility, which is supported by the ICTAS, the Macromolecules Innovation Institute, and the OVPRI in Virginia Tech. We thank Dr. N. Murthy Shanaiah (Department of Chemistry, Virginia Tech) for helping on the NMR experiments. This work was supported by the Jeffress Trust Awards (R.T. and H.X.), and the National Science Foundation (CHE-1807911 to R.T., and CBET-1845531 to H.X.).

## Author contributions

X.W., Y.H., H.X., and R.T. conceived the idea and designed experiments. X.W., X.X., Y.L., Z.H., R.T. performed the synthetic experiments and characterizations. Y.H., M.L. H.X., and R.T. carried out computational studies. X.W., Y.H., X.X., H.X., and R.T. analyzed the data, and wrote the manuscript.

## Competing interests

The authors declare no competing interests.
