## [Peer Review File · Nature Communications]

Bayesian-Optimization-Assisted Discovery of Stereoselective Aluminum Complexes for Ring-Opening Polymerization of Racemic LactideReviewers' Comments:

Reviewer #1:

Remarks to the Author:

This is an interesting and timely paper describing a Bayesian optimization approach to discover stereoselective lactide ring-opening polymerization catalysts. I think the manuscript provides new insights to the field and would be appreciated by the community. The paper can be accepted after following suggestions/comments are addressed.

- 1) Please describe the ML analysis methods more clearly. For instance, I could not find the information on how the authors obtained the validation set to perform hyperparameter tunings.
- 2) Have the authors tried other machine learning models other than GPR? Since the dataset is small, I assume that decision tree-based methods such as random forest regression would work better. If so, ones can use other optimization approach such as sequential model-based algorithm configuration (SMAC) instead of the Bayesian optimization to use RFR as a ML surrogate. In addition, SHAP analysis can more accurately be calculated using decision tree-based methods. The authors should make a comparison and comment on this to highlight the necessity of use of the Bayesian optimization approach.
- 3) I suggest to better describe the dataset. Although the dataset is uploaded, it is somewhat difficult to follow as it is a separate file. Please add some Figures by using some data visualization tools for better understanding of readers.
- 4) The interdependencies of the descriptors should be given as they could highly correlate each other.
- 5) The authors are encouraged to compare catalytic performance with literature value and/or state-of-the-art catalysts to clearly show the developed catalyst is unique and outstanding.

Reviewer #2:

Remarks to the Author:

In this manuscript, Hongliang Xin, Rong Tong, and coworkers report a Bayesian optimization workflow for machine learning to discover Al complexes for stereoselective ring-opening polymerization of racemic lactide. A fragmentation approach for the ligand design was used, and various DFT descriptors were calculated for training the model. The authors carried out multiple optimizations and evaluated the effectiveness of their prediction by synthesizing aluminum complexes and testing their experimental performance. The reported integration of computational methods with experimental validation provides a new avenue toward the design and development of selective Al complexes for ring opening polymerization.

Comments

1. Introduction: Authors should briefly explain how the stereochemistry of PLA influences its thermal and mechanical properties and why this is important. The introduction needs some background about it because this work focuses on the stereoselective lactide polymerization.
2. Introduction: Authors should explain why they specifically select salen-type Al complexes for this work in the introduction.
3. Line 147 – Figure 2b needs more discussion since this is the reason why authors use DFT-encoded descriptors for the rest of the studies.
4. Line 153: why use both Pr and Pm model since Pr and Pm are directly and mathematically related factors?
5. Figure 2 c and d: how was the convergence area selected? Also in figure d, it does not look like the two regions are overlapping.
6. Authors prepared 33 salen Al complexes that were proposed by their model. If there are any Al complexes that have not been reported before, they must be fully characterized by NMR spectroscopy (1H NMR values are included but no spectra are shown), elemental analysis, and X-ray crystallography. Showing 1H and 13C NMR spectra of pro-ligands is insufficient. If compounds were previously reported, need to clearly indicate literature references and include image of 1H NMR

spectrum for comparison.

7. Table 1 – The predicted Pm values of entries 3-6 are completely off. Authors need to discuss these mispredictions. Entry 6: the authors should comment on why the number-average molecular weight and molecular weight calculated from feed ratio and LA conversion are so different for A16C1B3. Also, entries 7 and 8 were included for comparison but not discussed. What is it to compare with?
8. Lines 243-246 – Authors only discuss the mean absolute error values for Pr values that are improved from 0.36 to 0.10. However, the mean absolute error values for Pm values are not improved and the value in the third round is even higher than that in the first and second rounds.
9. Figure 3e: The article lacks discussion on the difference between round 1, 2, and 3 optimization. How is it done? How were the catalysts selected for each round? Does the prediction get better with more rounds? Also, what is each point referring to?
10. Line 250 – Figure 3f is not relevant to authors' statement saying that the performance of their algorithm is efficient and capable of proposing valuable data points. Authors show the number of catalysts with Pm values higher than 0.8 and lower than 0.8. However, more important information is the accuracy of their algorithm. In SI Tables 3-8, some of proposed compounds have large discrepancies between the predicted and measured Pm or Pr values. With these data points, the performance does not look efficient and authors need to discuss these discrepancies even in the third round (e.g., Table 7, entry 4). Also, in the main text, the only sentence referring to this graph is "small size of the initial training dataset".
11. Figure 4a should be moved to the SI. It also needs a more detailed caption or explanation in the text. What does the positive and negative of SHAP value mean?
12. SI Figures 2-7 – Homodecoupled 1H NMR spectra of PLA need to show integration values of peaks.
13. SI Figure 11 should be included in the main text.
14. Supporting information: syntheses of ligands should include necessary characterizations (such as peaks in 1H NMR spectra) and yields of each ligand described. When more than one ligand is prepared applying one method, include the abovementioned information.
15. What is the point of including supplementary table 2 for the specific goal of this paper? Is the multistep synthesis included in the training and if so, how does it affect the predicted outcome? (also refer to main text line 182 to 185).
16. Supplementary table 4 entries 1, 3, 4, 6, 8 show predicted mean Pr that are all very different than measured Pr. What is the explanation?
17. SI: Some of the 1H NMR spectra show a significant amount of impurities, e.g., S-65 (a), S-75 (a), S-76 (a). Solvent signals should be labeled consistently.

Minor comments

1. Figure 1 is low resolution. Also, the flow chart is difficult to follow; the colors are similar and hard to read.
2. The word "symmetrical" in Figure 1 describing the ligand fragments is not accurate since R1 and R2 in A fragment are not always the same.
3. Figure 2 a and b should explain the error bar
4. Figure 2d: what does the blue band indicate?
5. Line 199 – "Pms" should be "Pm values".
6. Line 202: rac should be in italic.
7. GPC should be replaced by SEC throughout the paper and SI.
8. Figure 3a: A4C8B1 needs a better chemdraw, the bond is too short between the phenyl rings.
9. Figure 4b,c,d: the equations are hard to read and the color codes are confusing.
10. SI NMR spectra of ligands section – Each spectrum needs figure numbers.
11. Supplementary figures involving chemdraws of Al complexes need to use the same template and adjust the angles and distances accordingly. In the caption, include the type of homodecoupled NMR experiment (1H) and general conditions in which the spectra were taken.
12. For all NMR spectra: remove the title on the top left corner.

Reviewer #3:

Remarks to the Author:

The paper of Huang, Wang at al. describes the development of an algorithm for the identification of aluminum complexes able to work as stereoselective catalysts in the ring-opening polymerization of rac-lactide. Starting from stereoselective catalysts reported in the literature and employing their model the authors identify new stereoselective aluminum complexes and propose a correlation between two descriptors (%V_{bur} and HOMO energy) and the stereoselectivity of the complexes. Although the idea is interesting and could be useful for the development of new stereoselective catalysts, according to the reviewer the model is still in a preliminary stage and needs some improvements before it can actually achieve the desired ambitious purpose.

In fact, the new catalysts proposed by the model are very similar to those reported in the literature and do not bring significant improvements to the already reported results (in terms of stereoselectivity).

In addition, the model often fails in predicting stereoselectivity (see for example entries 3-6 of Table 1: four aluminum complexes predicted to be isoselective with a P_m greater than 0.80, and revealed to be heteroselective with P_r greater than 0.93).

One of the most significant critical points of the model is probably related to the lack of distinction between salen and salan ligands, overlooking the fact that salan ligands introduce new elements of chirality which may be crucial for the course of the reaction.

Moreover, the authors limit their investigations only to symmetric ligands, thus excluding both asymmetric salen and salan ligands and completely omitting salalen ligands which have proved to be interesting coordinating environments for stereoselective aluminum complexes.

Reviewer #1 (Remarks to the Author):

This is an interesting and timely paper describing a Bayesian optimization approach to discover stereoselective lactide ring-opening polymerization catalysts. I think the manuscript provides new insights to the field and would be appreciated by the community. The paper can be accepted after following suggestions/comments are addressed.

We thank for the reviewer's encouraging comments.

1) Please describe the ML analysis methods more clearly. For instance, I could not find the information on how the authors obtained the validation set to perform hyperparameter tunings.

We appreciate the reviewer's suggestion. We moved the implementation details of the ML model from the Supplementary Information to the main text. The mathematical details of ML methods can be found in the Supplementary Information sections S4.3 and S4.4, which also describe the principles of Gaussian process regression and the principles of Bayesian optimization, respectively.

We added below information about GPR and BO in S4.5:

“For Gaussian process regression (GPR), we use the Matern covariance kernel function. In the descriptor evaluation based on the 56 literature data points among one-hot encoding, density functional theory, electrotopological-state index and coulomb matrix descriptors, for each representation, feature decorrelation by removing features with Pearson correlation coefficient over 0.95 was carried out before training the ML model in order to reduce the chance of overfitting.”

“We did not use the cross-validation method to tune the GPR hyperparameters, therefore there is no validation set. Instead, due to the probabilistic nature of GPR, the hyperparameters are optimized by maximizing (with gradient-based optimizer Adam) the posterior likelihood function directly on the training set during training (see mathematical details in S4.3 in the Supplementary Information). In the 5-fold cross-validation test for descriptor comparison, there are also only training and test sets with the ratio of 4:1.”

“For Bayesian optimization, we use the Expected Improvement (EI) acquisition function.” ...

“For the exploration over the overall chemical space, in each iteration, we further select the model proposed new catalysts which are empirically estimated to be able to be synthesized within a small number of steps.”

2) Have the authors tried other machine learning models other than GPR? Since the dataset is small, I assume that decision tree-based methods such as random forest regression would work better. If so, ones can use other optimization approach such as sequential model-based algorithm configuration (SMAC) instead of the Bayesian optimization to use RFR as a ML surrogate. In addition, SHAP analysis can more accurately be calculated using decision tree-based methods. The authors should make a comparison and comment on this to highlight the necessity of use of the Bayesian optimization approach.

We thank for the reviewer’s suggestion. We used random forest regression (RFR) as the ML surrogate model and SMAC as the optimization method, compared to our current method (GPR as ML surrogate model and BO for the search). The results are documented in the section S4.9 in detail.

To summarize, we found that RFR and GPR exhibited similar prediction errors (in terms of both mean and standard deviation values) in a 5-fold cross-validation test (the results in the table in S4.9). In addition, for search or optimization performance, BO is more efficient than SMAC in terms of iterations numbers to achieve convergence in the search for the highest P_m and P_r values. The results are presented in the figures in S4.9.

We added below in the manuscript for the comparison:

“Additionally, our Bayesian optimization method was also found more efficient to reach convergence, compared to the sequential model-based algorithm configuration method⁶⁹ (see Supplementary Information S4.9).”

Furthermore, the SHAP analysis based on RFR (left) and our GPR method (right), as shown below, both indicate the importance of E_{HOMO} and V_{Bur} in determining P_m (or P_r) values. However, in terms of the range of SHAP value for each descriptor, the SHAP analysis based on RFR is only able to clearly quantify the correlation for the top-ranked descriptors, whereas the one based on our GPR algorithms can quantify more descriptors, which allows for the potential deep understanding of the nuance difference among structurally similar ligands.

We also added below in the manuscript:

“Notably, the SHAP analysis based on our GPR surrogate model allowed us to identify more descriptors compared to the analysis using the random forest regression⁷⁷ surrogate model (details see Supplementary Information S4.9).”

3) I suggest to better describe the dataset. Although the dataset is uploaded, it is somewhat difficult to follow as it is a separate file. Please add some Figures by using some data visualization tools for better understanding of readers.

We thank for the reviewer’s suggestion. As shown in supplementary Fig. 1, we collected 56 Al-complex molecules with their P_m or P_r values from the literature and used them as the dataset for descriptor selection, optimization benchmark with random search, and the initial training dataset to explore the overall chemical space. For Bayesian optimization, each molecule is originally represented by high-dimensional DFT descriptors, i.e., the combined vectors of DFT-calculated properties of A_m and B_nC_p fragments. The distribution of these 56 data points in the overall chemical space with a size of 576 can be visualized by projecting them into two dimensions using t-distributed stochastic neighbor embedding (t-sne) algorithm:

We added below discussions after Supplementary Fig. 1:

“The colored points are the data points in the initial training set, and the gray points are the rest of AI complexes in the overall chemical space. This distribution indicates two things: (1) the initial data points distribution is kind of localized, and the uncertainties in the regions without known P_m values nearby should be high and worth investigating; (2) Reasonable isoselective catalysts ($P_m > 0.5$) dominate in the initial training set, so uncertainties in the high P_r (low P_m) catalysts prediction would be high at the beginning of the optimization.”

For the three iterations of Bayesian optimization, all the data of model-proposed catalysts can be found in Supplementary Table 3, 5, 7 for the search of high P_m catalysts and Supplementary Table 4, 6, 8 for the search for high P_r catalysts. The data includes: ligand code, model predicted mean and variance of P_m/P_r value, measured P_m/P_r value, experimental conditions, LA monomer conversion, number-average molecular weight, molecular weight calculated from the feed ratio and monomer conversion, and molecular weight distribution. We note that these data are discrete, because our algorithm provided proposed ligands with predicted P_m/P_r values as the output. The only way to visualize the ligands proposed by the model is to draw chemical structures. To visualize the P_m/P_r value difference between the experiments and model, we have provided Figure 3e. To visualize the model prediction performance, we have provided Figure 3f, and the stereoselective ligands with high P_m/P_r values were shown in Figure 2a.

To this end, we have uploaded the dataset together with all the scripts and README file to our Github repository: <https://github.com/hlxin/bayespoly> For example, in the dataset named “experiment_index.csv”, as shown below. YR3_code and R1R2_code refer to the specific fragment of BnCp and Am, respectively in that AI-complex molecule. Temp_code and Ratio_code are constants referring to temperature and monomer/catalysts feed ratio in the reaction, both of which can be ignored because we did not consider them in the model training and optimization. In the feature set which is fed into the machine learning model, each of YR3_code and R1R2_code

is expanded in terms of the descriptor vectors depending on which descriptor type is chosen, i.e. DFT/Mordred/CM/EI/OHE (<https://github.com/hlxin/bayespoly/tree/main/BO/data/AI>).

entry	YR3_code	R1R2_code	Temp_code	Ratio_code	yield	catalyst
0	31	15	70	100	0.97	A15Y3B1
1	31	3	70	100	0.93	A3Y3B1
2	51	3	70	100	0.93	A3Y5B1

4) The interdependencies of the descriptors should be given as they could highly correlate each other.

We actually performed feature decorrelation that removed highly correlated features (Pearson correlation coefficient > 0.95) as one step of data preprocessing before the model training. In the section S4.10, to elucidate the interdependencies of descriptors, we added below discussions:

“With DFT descriptors, from the Pearson correlation matrix, the highly correlated feature pairs are extracted and tabulated below (x: A_m fragment, y: B_nC_p fragment). Below are the relationships with high correlation: (1) For both A_m and B_nC_p , the number of atoms is positively correlated with all four thermodynamic energy corrections. This is obvious because more atoms mean higher degrees of freedom and free energy correction per degree of freedom is a constant. (2) For both A_m and B_nC_p , the four thermodynamic energy corrections are correlated to each other. The reason is the same as the first one, i.e. all the correction terms are linearly dependent on the number of atoms so they themselves are also linearly correlated to each other. (3) For both A_m and B_nC_p , the four thermodynamic energies, i.e. self-consistent field energy (DFT energy), ZPE (zero-point energy), E, H, G are almost the same. This is because the thermodynamic energy corrections ($\sim 10^{-1}$) added to the self-consistent field energies are so tiny compared to the self-consistent field energies ($\sim 10^3$) themselves. (4) For A_m , the LUMO energy is negatively correlated with electronegativity. This is because the electronegativity here is defined as the negative value of the average of HOMO and LUMO energies and these two energies are kind of positively correlated to each other. (5) For B_nC_p , the number of atoms is negatively correlated with all the four energies. The reason is that various B_nC_p mainly differs in the number of carbon atoms and more carbon atoms lead to more C-C bonds and lower formation energies. (6) For B_nC_p , the electronic spatial extension is negatively correlated with all four energies. The reason is that the more atoms included, the larger the molecular fragment can be, and further out, the electronic density can extend with significant probability, i.e., higher electronic spatial extension. And the number of atoms is negatively correlated with the total energy. (7) For B_nC_p , the mean

reduced mass is negatively correlated to the mean frequency. This can be understood in a simple harmonic oscillation model where the vibrational frequency is expressed as:

$$\nu = \frac{1}{2\pi} \sqrt{\frac{k}{\mu}},$$

where μ is the reduced mass of the system.”

We also included the table to show the interdependencies of descriptors and their Pearson correlation coefficient in S4.10.

5) The authors are encouraged to compare catalytic performance with literature value and/or state-of-the-art catalysts to clearly show the developed catalyst is unique and outstanding.

We thank for the reviewer’s suggestion. Our Table 1 actually made the comparison of some of our best catalysts to the literature ones. To further clarify, for high- P_m catalysts, we added:

“The Al complexes with $A_{11}C_3B_1$ and $A_{11}C_2B_1$ both afford stereoblock copolymers with P_m values over 0.92 (representative homodecoupled 1H NMR of the α -methine region in PLA prepared using $(A_{11}C_3B_1)Al$ complex in Fig. 3c; NMR spectra of the PLA synthesized by other Al complexes in Supplementary Figs. 2-7), and high monomer conversions > 95% over 12 hours ($[rac-LA]/[Al] = 100/1$, Table 1, entries 1-2), which exhibited slightly better isoselectivity control compared with the previously reported $(A_3C_3B_1)Al$ complex⁴⁷ (Fig. 3c; Table 1, entries 1 versus 7).”

For high- P_r catalysts, we added

“All of these four Al catalysts showed markedly improved stereocontrol in the preparation of highly heterotactic PLA, when compared to the previously reported $(A_6C_1B_3)Al$ complex⁴⁵ (Table 1, entries 3-6 versus 8).”

We also mentioned that our catalyst ligands can be prepared within 3 steps, which significantly save the synthesis time and allow for future scale-up. To emphasize that, we added:

“We focused on ligands that could be prepared in no more than three steps because we prioritized accelerated catalyst discovery over exploration of whole chemical spaces, and time presents a substantial cost (note that some synthetic-demanding ligands, e.g., $A_{15}C_3B_1$ —even its Al complex showing excellent isoselectivity⁴⁷—requiring 5 steps to prepare, would not be considered owing to time and materials limitations).”

Here the $(A_{15}C_3B_1)Al$ complex had the best P_m ($P_m = 0.97$) in the literature, slightly higher than our $(A_{11}C_3B_1)Al$ complex ($P_m = 0.95$). Nevertheless, such catalyst is synthetically demanding, requiring 5 steps to prepare the ligands, and is not practical for large-scale productions. Our newly-identified stereoselective catalysts do not have such highly-synthetically-demanding problems.

Additionally, we note that some of our complexes (e.g., high- P_r catalysts based on A_5) are unlikely to be identified based on the literature data or experiences as only one literature data point is available and the value is moderate:

“...the two heteroselective ligands based on A_5 , which had only one P_m data point ($P_m = 0.76$) in the literature,⁵³ would not likely have been investigated on a trial-and-error, screening, or intuition-guided study.”

Moreover, our data-driven workflow using DFT-based descriptors could uncover and quantify structural factors impacting the stereoselective polymerization results, as shown in Figure 4. Given the initial small-sized training data (56 data points), and most of them are isoselective Al complexes (40 complexes with $P_{ms} > 0.5$), the efficient identification of five heteroselective catalysts with $P_{rs} > 0.5$ clearly demonstrates the state-of-art of machine learning techniques in accelerating polymerization catalyst development.

Reviewer #2 (Remarks to the Author):

In this manuscript, Hongliang Xin, Rong Tong, and coworkers report a Bayesian optimization workflow for machine learning to discover Al complexes for stereoselective ring-opening polymerization of racemic lactide. A fragmentation approach for the ligand design was used, and various DFT descriptors were calculated for training the model. The authors carried out multiple optimizations and evaluated the effectiveness of their prediction by synthesizing aluminum complexes and testing their experimental performance. The reported integration of computational methods with experimental validation provides a new avenue toward the design and development of selective Al complexes for ring opening polymerization.

We thank for the reviewer's positive comments.

Comments

1. Introduction: Authors should briefly explain how the stereochemistry of PLA influences its thermal and mechanical properties and why this is important. The introduction needs some background about it because this work focuses on the stereoselective lactide polymerization.

We thank for the reviewer's suggestion. We added below in the introduction:

“For example, poly(lactic acid) (PLA) with stereoregular microstructures, e.g., isotactic structures, is crystalline, and have improved thermal properties (e.g., specific melting temperature) and mechanical properties (e.g., higher tensile modulus) than the atactic PLA.^{12, 14}”

We would like to mention that we did characterize thermal and mechanical properties of PLAs with specific microstructures using our newly developed Al catalysts, as shown in Figure 3d and Supplementary Table 9.

2. Introduction: Authors should explain why they specifically select salen-type Al complexes for this work in the introduction.

We added below in the first section “Benchmarking the machine-learning algorithms”:

“We focused on salen- and salan-type Al complexes because they are the most frequently studied metal complexes for the ROP of *rac*-LA,^{14, 41, 46} which might provide reasonably sufficient data points to initiate machine learning.”

We note that although other metals might also have many ligands, however, most them have too many varied sites on the ligands and are difficult to categorize, or many have asymmetric structures

which is synthetic demanding (more than 3 steps to prepare the ligands). We added above discussions under Supplementary Table 1 (also see comments on reviewer 3' last two questions).

3. Line 147 – Figure 2b needs more discussion since this is the reason why authors use DFT-encoded descriptors for the rest of the studies.

We discussed Figure 2b in our manuscript: “Notably, DFT-encoded descriptors, whose parity plot between GPR-predicted and measure P_m values shown in Fig. 2b, could provide rich chemical information with insights for reaction mechanism studies. Therefore, we carried out the remainder of the studies using DFT-encoded descriptors.” The reason for selecting DFT is also because of the reason mentioned below in the manuscript: “We found that consistent regression performance—lowest mean errors and standard deviations—could be achieved using the datasets generated by one-hot encoding, Mordred and DFT (Fig. 2a; details in Supplementary Information S4.5)” not just Figure 2b. Note that the other two types of descriptors (OHE and Mordred) could not provide mechanistic insights as we discussed in Figure 4.

4. Line 153: why use both P_r and P_m model since P_r and P_m are directly and mathematically related factors?

It is true that P_m and P_r are closely related to each other, i.e. $P_m + P_r = 1$. So only a single surrogate model is needed to be trained to predict P_m and P_r . However, since there are two optimization directions, i.e. high P_m /low P_r , and low P_m /high P_r , we must need two Bayesian optimization procedures, one for maximizing P_m and the other for maximizing P_r (minimizing P_m). These two processes differ in the definition of the acquisition function, which is used to select promising points in each iteration. In process of maximizing P_m , the acquisition function is increased as the mean and variance values of predicted P_m increase. On the other hand, in the process of maximizing P_r , the acquisition function is increased as the mean and variance values of predicted P_r increase (P_m decrease). Therefore, one Bayesian optimization procedure cannot meet for the optimization of both directions, and we need two.

We added below in the main text:

“Two Bayesian optimization models were built for P_m and P_r , respectively, owing to the two distinctive Bayesian optimization directions.”

5. Figure 2 c and d: how was the convergence area selected? Also in figure d, it does not look like the two regions are overlapping.

In Figure 2c-d, in each iteration, we carried out 10 independent optimization experiments to search for the maximum P_m (or P_r) values, using either Bayesian optimization or random search method. Through the iteration, the model is expected to be refined to quickly identify the maximal points. When 10 experiments all find the optimal points, we say that the convergence of that optimization method is reached, i.e., the standard deviation becomes zero. In both P_m and P_r cases, eventually the blue band diminishes (via Bayesian optimization) and the red one does not (via random search method), which means that Bayesian optimization method gets converged while random search

does not. Note that the band is the standard deviation range, not convergence area. The iteration number that the model reaches convergence, i.e. standard deviation becomes zero, is highlighted by the arrow.

In Figure 2 legend, to explain the concept of convergence, we added below:

“...Each optimization process was independently repeated for 10 runs (12 iterations per run). Data are shown as the mean value with the standard deviation (bandwidth) of the highest P_m or P_r observed in each iteration. The Bayesian optimization curves in (c) and (d) both achieved convergence (i.e. the blue band diminished, pointed with the arrows) within 7 rounds. In contrast, the random search process exhibited large standard deviations (i.e., the red band never diminished) and failed to converge within 12-round optimization.”

6. Authors prepared 33 salen Al complexes that were proposed by their model. If there are any Al complexes that have not been reported before, they must be fully characterized by NMR spectroscopy (^1H NMR values are included but no spectra are shown), elemental analysis, and X-ray crystallography. Showing ^1H and ^{13}C NMR spectra of pro-ligands is insufficient. If compounds were previously reported, need to clearly indicate literature references and include image of ^1H NMR spectrum for comparison.

We added ^1H and ^{13}C NMR spectra for most Al complexes. We note that some Al complexes had low solubility in all NMR solvents (we tried CDCl_3 , toluene- d_8 and THF- d_8), and we have to use high-temperature NMR (85 °C) to acquire ^1H NMR spectra (we could not acquire ^{13}C NMR at such high temperature due to the limitation of the facility). We note that all these room-temperature-less-soluble Al complexes were washed by ether and hexane to remove the ligands (all of their ligands are soluble in toluene and ether). To further confirm the structure, we performed high-resolution MS analysis. Notably, all room-temperature-less-soluble Al complexes with broad peaks in NMR spectra did not present any stereoselectivity control towards the ROP of *rac*-LA, which did not affect our current conclusions. We would like to mention that the elemental analysis and X-ray crystallography in our institute are currently not accessible due to the staff shortage and lack of maintenance during the pandemic. Since our current work focus on the development of the machine learning framework for catalyst discovery, our currently available NMR and MS data supported our successful preparation and discovery of Al complexes guided by Bayesian optimization models.

7. Table 1 – The predicted P_m values of entries 3-6 are completely off. Authors need to discuss these mispredictions. Entry 6: the authors should comment on why the number-average molecular weight and molecular weight calculated from feed ratio and LA conversion are so different for A16C1B3. Also, entries 7 and 8 were included for comparison but not discussed. What is it to compare with?

We thank that the reviewer pointed out typos in Table 1. The original values in the column of “Predicted mean P_m ” of entries 3-6 are actually numbers of “Predicted mean P_r ”. We have corrected these in the manuscript. Actually, we had the correct numbers in the Supplementary

Information and in the Figure 3a-b when we submitted the manuscript, but we did not convert them in the entries 3-6 in Table 1.

For entry 6, the obtained polymer's M_n is higher than the calculated molecular weight. We doubt that at 70 °C reaction temperature, moderate chain transfer might occur. Nevertheless, the molecular weight distribution was 1.08, suggesting no significant transesterification happening that often causes large distribution. We added above discussion after Supplementary Table 8, whose entry 2 corresponds to entry 6 in Table 1.

Entries 7 and 8 are reported complexes and we used to benchmark for comparison. We added below discussions to compare with our newly prepared catalysts:

“The Al complexes with $A_{11}C_3B_1$ and $A_{11}C_2B_1$ both afforded stereoblock copolymers with P_m values over 0.92 (representative homodecoupled 1H NMR of the α -methine region in PLA prepared using $(A_{11}C_3B_1)Al$ complex in Fig. 3c; NMR spectra of the PLA synthesized by other Al complexes in Supplementary Figs. 2-7), and high monomer conversions > 95% over 12 hours ($[rac-LA]/[Al] = 100/1$, Table 1, entries 1-2), which exhibited slightly better isoselectivity control compared with the previously reported $(A_3C_3B_1)Al$ complex⁴⁷ (Fig. 3c; Table 1, entries 1 versus 7).”

and:

“All of these four Al catalysts showed markedly improved stereocontrol in the preparation of highly heterotactic PLA, when compared to the previously reported $(A_6C_1B_3)Al$ complex⁴⁵ (Table 1, entries 3-6 versus 8).”

8. Lines 243-246 – Authors only discuss the mean absolute error values for P_r values that are improved from 0.36 to 0.10. However, the mean absolute error values for P_m values are not improved and the value in the third round is even higher than that in the first and second rounds.

First, the mean absolute errors for P_m values in the first round, second round, and third round do not have significant differences. For the optimization of P_m , the P value between first and third round is 0.60, and the P value between first and second rounds is 0.43 (all by Mann-Whitney U test). Thus, it is fair to say the mean absolute errors for P_m values did not have significant improvement compared to that process in the prediction of P_r . Here are the reasons:

(1) In our initial 56-catalyst training dataset, there are 47 catalysts with P_m over 0.5, and only 9 with P_r over 0.5. Therefore, it is much more difficult to optimize P_r than P_m at the beginning of Bayesian optimization process. Such limited information in the high- P_r catalysts led to the high error in the first round of prediction on catalysts potentially with high P_r . The high uncertainty in the highly unexplored high- P_r catalyst region quickly decreased as several high- P_r (>0.8) were identified, and the subsequent addition of such data into the training dataset further refined the model, thereby resulting in a significant decrease in the error.

(2) On the other side, for the P_m optimization, the sufficient number of reasonable P_m -value catalysts in the initial training dataset (47 catalysts having $P_m > 0.5$, and 18 of them having $P_m \geq 0.8$) made the uncertainty of P_m optimization process much lower compared to that in the P_r optimization.

(3) Additionally, many factors—such as experimental measurements (errors in P_r / P_m measurements) and errors in literature data (e.g., some literature data was found reportedly higher when we tried to repeat, and we could not repeat all reported catalysts) — would bring the uncertainty into the system, which can be hard to shrink down even after a large number of iterations of Bayesian optimization. This uncertainty could account for a large portion in the prediction error after just three iterations.

(4) Notably, even after three iterations of surrogate model training, the final training dataset size (86) compared to the overall chemical space size (576) is still relatively small and the unexplored regions of the chemical space still have high prediction uncertainty. Theoretically, more iterations can lead to an increase in prediction accuracy. However, our whole framework aims to accelerate the discovery of highly stereoselective complexes under the guidance of the machine learning model using minimal resources. To achieve this goal, we utilize highly efficient BO process and introduce the concept of synthetic scales to minimize both time and reagent costs in the catalyst synthesis, thereby accelerating the discovery of high-performance catalysts ($P_r / P_m > 0.8$). We would like to emphasize that repetitive iterations that explore the whole chemical spaces—though eventually minimizing the errors in the model—is not the aim of our framework, and such labor-intensive endless iterations are also not economical and translatable to discover / optimize other catalytic systems.

To further elucidate such confusion, we added below into the main text:

“We reason that the limited heteroselective catalyst information—only 9 of 56 complexes having P_r values over 0.5—at the beginning of the search could contribute to the high prediction error in the first round. Nevertheless, such high uncertainty in the unexplored high- P_r -value region in the chemical space quickly decreased when several high- P_r -value complexes were identified and appended into the training set to refine the model, thereby significantly decreasing the prediction errors in the subsequent rounds.”

The discussion about the prediction errors of P_m was added after Supplementary Table 7.

9. Figure 3e: The article lacks discussion on the difference between round 1, 2, and 3 optimization. How is it done? How were the catalysts selected for each round? Does the prediction get better with more rounds? Also, what is each point referring to?

In our original manuscript, we have described the whole process of selecting catalysts in each round:

“The model subsequently proposed ligands potentially having high P_m or P_r values. The predicted points were ranked by the acquisition function, and we selected the top-ranked data points to prepare ligands and verify their stereoselectivities in ROP of *rac*-LA. To circumvent prohibitively multistep, reagent- and time-consuming syntheses, we assigned each substituent a metric called “expected synthetic difficulty,” which was based on the sum of the expected number of steps required to synthesize A_m and B_nC_p and thus to build the whole Al complex (Supplementary Table 2). We focused on ligands that could be prepared in no more than three

steps because we prioritized accelerated catalyst discovery over exploration of whole chemical spaces.”

In other words, the catalysts in each iteration are proposed through two steps: first, we choose the ones with high values of acquisition function, which is an optimality metric built on the surrogate model outputs (predicted mean and variance of P_m or P_r values) to evaluate how promising each point is in the whole chemical space. Next, from such proposed points, we select ones that can be easily synthesized within three steps.

We also mentioned:

“The experimental data points obtained for the proposed ligands in each round were appended to the dataset to refine the model for next-round prediction (Fig. 1a).”

The experimental results were put back into the model to finish one round and start the next round. We also discussed the difference among the round 1 and the next two rounds in our original manuscript (in short, we removed the temperature and reaction conditions in the rounds 2 and 3):

“In our first-round modeling, we included polymerization reaction parameters such as temperature and monomer concentration, but these parameters were found to be less relevant to the results of predicted points compared with the catalyst descriptors. This is because the chemical structures of the ligands largely determined the stereoselectivity in polymerization.”

To explain “the points”, we added below in the main text:

“we selected the top-ranked data points (i.e. the most promising ligands proposed by the model) to prepare ligands...”

10. Line 250 – Figure 3f is not relevant to authors’ statement saying that the performance of their algorithm is efficient and capable of proposing valuable data points. Authors show the number of catalysts with P_m values higher than 0.8 and lower than 0.8. However, more important information is the accuracy of their algorithm. In SI Tables 3-8, some of proposed compounds have large discrepancies between the predicted and measured P_m or P_r values. With these data points, the performance does not look efficient and authors need to discuss these discrepancies even in the third round (e.g., Table 7, entry 4). Also, in the main text, the only sentence referring to this graph is “small size of the initial training dataset”.

We added below to better explain Figure 3f:

“The portion of high-performance stereoselective Al catalysts (P_m or $P_r > 0.8$) discovered in the third iteration is higher than the first and second iterations, suggesting an improved search efficiency of our model (Figure 3f).”

We also would like to clarify that the sentence “small size of the initial training dataset” was meant to describe the small size of the initial training dataset and fair prediction accuracy based on it. We removed “Figure 3f” from that sentence.

For regression accuracy, we admit that the improvement of model prediction accuracy is limited due to the small number of new Al-complexes being added into the training set to refine the model. The unexplored regions of the chemical space still have high prediction uncertainty, and it is reasonable to still observe errors even in the third iteration. As we mentioned throughout the

manuscript, we prioritize the accelerated catalyst discovery over the exploration of whole chemical spaces, and time presents a substantial cost in the process. The refinement of our BO model to negligible errors is not the goal of this study. Considering the time and experimental cost, we didn't carry out further iteration by continuing synthesizing more ligands to refine the model. Note that the newly added data points have already allowed us to perform SHAP analysis to understand the relationship among catalysts' descriptors and their performances, as shown in Figure 4. In addition, as we mentioned in the question # 8, many realistic factors affect the experimental measurement, the ground truth of P_m/P_r values have uncertainty that is hard to shrink down or diminish even after a large number of iterations of Bayesian optimization. Most importantly, the reason why we perform multiple iterations is not trying to refine the model to make it more accurate, but mainly aiming to efficiently find more high-performance catalysts (P_m or $P_r > 0.8$). Although the accuracy improvement is limited, we believe that we have already achieved our goal of finding more high-performance catalysts guided by Bayesian optimization.

Part of the discussion about the uncertainty was added after Supplementary Table 7.

11. Figure 4a should be moved to the SI. It also needs a more detailed caption or explanation in the text. What does the positive and negative of SHAP value mean?

Figure 4a is informative in terms of mechanistic understandings from BO models, and we extracted all the important mechanistic descriptors for the attribution analysis of the BO model. Such information and knowledge are so important, and such SHAP analysis strategy has rarely been applied in using the BO model for optimizing and understanding organic chemical reactions, and we believe that it should be kept in the main text.

Positive/negative SHAP value refers to positive/negative correlation between the measured P_m value and the corresponding descriptor. We added below in the main text to explain the SHAP value:

“The positive and negative SHAP value refers to positive and negative correlation, respectively, between the measured P_m value and the corresponding descriptor. The larger the absolute value, the stronger the correlation is.”

12. SI Figures 2-7 – Homodecoupled ^1H NMR spectra of PLA need to show integration values of peaks.

We provided the integration area values for all homodecoupled ^1H NMR spectra used for the calculations of P_m values. For calculating P_r values, we found using ^{13}C NMR spectra would give more accurate results. Detailed rationale and justifications were discussed in Supplementary Figs. 2 and 3.

13. SI Figure 11 should be included in the main text.

We have mentioned in the main text “such DFT computation of transition states was costly, and was provided as an afterthought for the reaction development, especially in this case where multiple conformers needed to be examined.” The reason that we performed the DFT calculations

was to show that even the mechanism of ring-opening of LA has already been proposed and understood, using the mechanism-based transition state DFT calculation to search for catalysts can be costly (multiple intermediate states calculation), and also could not guide the search process (cannot using such calculation results to propose/predict ligands). Therefore, the discussion of DFT mechanism only served as an afterthought, and runs counter toward our efficient BO-based framework, which should not be included in the main text (the mechanism is also known in the literature). Solely relying on the DFT-based transition state calculation could be a dreadful strategy for the discovery of high-performance catalysts.

14. Supporting information: syntheses of ligands should include necessary characterizations (such as peaks in ^1H NMR spectra) and yields of each ligand described. When more than one ligand is prepared applying one method, include the abovementioned information.

We thank for the reviewer's suggestion and additional synthetic information such as separation/purification methods, color/states of the product and yields were added.

15. What is the point of including supplementary table 2 for the specific goal of this paper? Is the multistep synthesis included in the training and if so, how does it affect the predicted outcome? (also refer to main text line 182 to 185).

Conventional catalyst screening method is both time-consuming and materials-extensive, and relies on the chemists' experience. Here we would like to use data science tools to accelerate the discovery process. In our manuscript, we mentioned "To circumvent prohibitively multistep, reagent- and time-consuming syntheses..." and "because we prioritized accelerated catalyst discovery over exploration of whole chemical spaces, and time presents a substantial cost", thereby adding such expected synthetic difficulty based on synthetic steps needed to prepare a ligand becomes indispensable to our goal. In other words, we would like to prepare ligands within three steps to increase the discovery process, which would also be useful for future scale-up.

Notably, the consideration of the number of synthesis steps is not directly enumerated as the model input, but it could affect the subsequent model training and prediction because it partially determines (besides the acquisition function from the model output) the new points we selected in each iteration and thus the updated training dataset. We don't know how different the prediction would be if we did not consider it, because the synthesis requiring many steps has a low success rate and prevents us from obtaining enough new data to update the model within a reasonable amount of time, which might impede the success of this project.

16. Supplementary table 4 entries 1, 3, 4, 6, 8 show predicted mean P_r that are all very different than measured P_r . What is the explanation?

As mentioned in our previous responses to the reviewer's comment # 8, there are 47 catalysts with P_m over 0.5, and only 9 catalysts having P_r over 0.5 in our initial training dataset. Therefore, optimization of P_r was considerably difficult comparing to optimizing P_m in our first round BO prediction. Given the limited information of heteroselective catalysts in the first round, the errors

can be large in the prediction. In addition, in our first-round prediction, the descriptors of temperature and monomer feed ratios were included in the model, which unnecessarily enlarged the chemical space for prediction, thereby further increased the uncertainty and led to high prediction errors. After realizing the low relevance of these descriptors to the P_m / P_r value prediction, we removed them in the subsequent iterations, which could also contribute to the decreased errors in later P_r value prediction.

We added above discussions after Supplementary Table 4.

17. SI: Some of the ^1H NMR spectra show a significant amount of impurities, e.g., S-65 (a), S-75 (a), S-76 (a). Solvent signals should be labeled consistently.

We thank the reviewer's suggestions and labeled the solvent signals in the NMR spectra. In our revised SI, we noted that Supplementary Figures S13-S45 are ligands' NMR spectra, and Supplementary Figures S55-S87 are Al complexes' NMR spectra.

Minor comments:

1. Figure 1 is low resolution. Also, the flow chart is difficult to follow; the colors are similar and hard to read.

We actually uploaded the high-resolution figures (300 dpi) separately when we submitted our manuscript. The low resolution is because the pdf converted by the nature manuscript system lowered the figure resolution, which may also blur the colors. This time we also upload a separate main-text pdf file with high resolution figures, along with the high-resolution image tif files. For colors in figures, we use blue, light red, light green, yellow, dark red etc. in the figure, which should be distinguishable.

2. The word "symmetrical" in Figure 1 describing the ligand fragments is not accurate since R1 and R2 in A fragment are not always the same.

In Figure 1, the word "symmetrical" refer to the whole ligand not the A_m fragment. Even R_1 and R_2 are different, however, the left side of the ligand has the same R_1 and R_2 as the right side.

3. Figure 2 a and b should explain the error bar

To explain the error bars, for Figure 2a we added "The error bars are the standard deviations of prediction errors in the 5-fold cross validations." For Figure 2b we added "The error bars are the predicted standard deviation values."

4. Figure 2d: what does the blue band indicate?

In Figure 2 legend, we added "Data are shown as the mean value with the standard deviation (band width) of the highest P_m or P_r observed up to each iteration." The blue band is the range of \pm standard deviations.

5. Line 199 – "Pms" should be "Pm values".

We changed accordingly to “ P_m values”.

6. Line 202: rac should be in italic.

We changed accordingly.

7. GPC should be replaced by SEC throughout the paper and SI.

We changed accordingly.

8. Figure 3a: A4C8B1 needs a better chemdraw, the bond is too short between the phenyl rings.

We replaced Figure 3 with newly plotted A₄C₈B₁ structure.

9. Figure 4b,c,d: the equations are hard to read and the color codes are confusing.

We thank the reviewer for pointing that out. We moved the equations on the top of the figures and enlarged the fonts for readers to read. For the colors, in the figure legend, we added “In the equations in (c-e), the descriptors of the A_m fragment are highlighted in purple, and those of the B_nC_p fragment in green, and residue numbers in pink.”

10. SI NMR spectra of ligands section – Each spectrum needs figure numbers.

We added figure numbers for the ligands accordingly. We noted that Supplementary Figures S13-S45 are ligands’ NMR spectra, and Supplementary Figures S55-S87 are Al complexes’ NMR spectra.

11. Supplementary figures involving chemdraws of Al complexes need to use the same template and adjust the angles and distances accordingly. In the caption, include the type of homodecoupled NMR experiment (¹H) and general conditions in which the spectra were taken.

We thank for the reviewer’s suggestion and re-drew many Al complexes structures. We also noted “¹H homodecoupled NMR spectra” and added general conditions (frequency and solvent) for the spectra.

12. For all NMR spectra: remove the title on the top left corner.

We changed accordingly.

Reviewer #3 (Remarks to the Author):

The paper of Huang, Wang at al. describes the development of an algorithm for the identification of aluminum complexes able to work as stereoselective catalysts in the ring-opening polymerization of rac-lactide. Starting from stereoselective catalysts reported in the literature and employing their model the authors identify new stereoselective aluminum complexes and propose a correlation between two descriptors (%V_{bur} and HOMO energy) and the stereoselectivity of the complexes.

We thank for the reviewer's comments.

Although the idea is interesting and could be useful for the development of new stereoselective catalysts, according to the reviewer the model is still in a preliminary stage and needs some improvements before it can actually achieve the desired ambitious purpose. In fact, the new catalysts proposed by the model are very similar to those reported in the literature and do not bring significant improvements to the already reported results (in terms of stereoselectivity).

We disagree the reviewer's comments "the new catalysts proposed by the model are very similar to those reported in the literature and do not bring significant improvements to the already reported results (in terms of stereoselectivity)". In Figure 3c, we showed that our newly discovered catalyst outperformed the literature-reported catalysts. To clarify that point, for high- P_m catalysts, we compared with the reported ($A_{11}C_3B_1$)Al complex and added:

"The Al complexes with $A_{11}C_3B_1$ and $A_{11}C_2B_1$ both afford stereoblock copolymers with P_m values over 0.92 (representative homodecoupled 1H NMR of the α -methine region in PLA prepared using ($A_{11}C_3B_1$)Al complex in Fig. 3c; NMR spectra of the PLA synthesized by other Al complexes in Supplementary Figs. 2-7), and high monomer conversions > 95% over 12 hours ($[rac-LA]/[Al] = 100/1$, Table 1, entries 1-2), which exhibited slightly better isoselectivity control compared with the previously reported ($A_3C_3B_1$)Al complex⁴⁷ (Fig. 3c; Table 1, entries 1 versus 7)."

For our high- P_r catalysts, we found Gibson's catalysts had a lower conversion % of LA (76% for 12 hours at 70 °C) and lower P_r (0.89 instead of 0.96) through head-to-head comparison with our newly discovered complexes (all above 0.93). We noted:

"All of these four Al catalysts showed markedly improved stereocontrol in the preparation of highly heterotactic PLA, when compared to the previously reported ($A_6C_1B_3$)Al complex⁴⁵ (Table 1, entries 3-6 versus 8)."

We also mentioned that our catalyst ligands can be prepared within 3 steps, which significantly save the synthesis time and allow for future scale-up. To emphasize that, we added:

"We focused on ligands that could be prepared in no more than three steps because we prioritized accelerated catalyst discovery over exploration of whole chemical spaces, and time presents a substantial cost (note that some synthetic-demanding ligands, e.g., $A_{15}C_3B_1$ —even its Al complex showing excellent isoselectivity⁴⁷—requiring 5 steps to prepare, would not be considered owing to time and materials limitations)."

Here the ($A_{15}C_3B_1$)Al complex had the best P_m ($P_m = 0.97$) in the literature, slightly higher than our ($A_{11}C_3B_1$)Al complex ($P_m = 0.95$). Nevertheless, such catalyst is synthetically demanding, requiring 5 steps to prepare the ligand, and impractical for large-scale productions. Our newly-identified stereoselective catalysts do not have such highly-synthetically-demanding problems. PLA with different microstructures can therefore be prepared with decent molecular weights over 40 kDa in gram scales for mechanical properties tests, as shown in Figure 3d.

Additionally, we note that some of our complexes (e.g., high- P_r catalysts based on A_5) are unlikely to be identified based on the literature data or experiences as only one literature data point is available and the value is moderate:

“...the two heteroselective ligands based on A_5 , which had only one P_m data point ($P_m = 0.76$) in the literature,⁵³ would not likely have been investigated on a trial-and-error, screening, or intuition-guided study.”

Moreover, our data-driven workflow using DFT-based descriptors could uncover and quantify structural factors impacting the stereoselective polymerization results, as shown in Figure 4. Given the initial small-sized training data (56 data points), and most of them are isoselective Al complexes (40 complexes with $P_{ms} > 0.5$), the efficient identification of five heteroselective catalysts with $P_r > 0.5$ clearly demonstrates the state-of-art of machine learning techniques in accelerating polymerization catalyst development.

In addition, the model often fails in predicting stereoselectivity (see for example entries 3-6 of Table 1: four aluminum complexes predicted to be isoselective with a P_m greater than 0.80, and revealed to be heteroselective with P_r greater than 0.93).

We apologized for the typos in Table 1: the original values in the column of “Predicted mean P_m ” of entries 3-6 are actually numbers of “Predicted mean P_r ”. We have corrected these in the manuscript. Actually, we had the correct numbers in the Supplementary Information and in Figure 3a-b when we submitted the manuscript, but we did not convert them in the entries 3-6 in Table 1. Therefore, this model did not “often fails in predicting stereoselectivity”.

One of the most significant critical points of the model is probably related to the lack of distinction between salen and salan ligands, overlooking the fact that salan ligands introduce new elements of chirality which may be crucial for the course of the reaction.

We disagree the reviewer’s statement “the model is probably related to the lack of distinction between salen and salan ligands”. In our model, salen ligands correspond to $A_mC_pB_1$ and salan ligands correspond to $A_mC_pB_2$ and $A_mC_pB_3$ ligands. To further clarify that, we revise the main text to use “salen- and salan-type Al complexes”.

We are fully aware of chiral centers could be introduced to both salen and salan ligands. Al complexes bearing enantiopure binaphthyl (*Macromol. Chem. Phys.* **1996**, *197*, 2627; *J. Am. Chem. Soc.* **2002**, *124*, 1316) or bipyrrrolidine ligands (*Angew. Chem., Int. Ed.*, 2015, **54**, 14858; *Chem. Sci.*, 2015, **6**, 5034; *Angew. Chem., Int. Ed.*, 2019, **58**, 14679; *J. Am. Chem. Soc.* 2022, **144**, 2004) have been shown to mediate stereoselective ROP via enantiomorphic site control. A few reasons listed below lead us not to adapt these interesting ligands in our initial establishment of adopting Bayesian optimization for stereoselective catalyst development:

(1) When introducing these chiral ligands, different stereoisomers of ligands have to be studied, namely the (R,R), (S,S), the meso (R,S) forms, and the racemic form ($(R,R) + (S,S)$). It remains difficult now to establish machine learning models using DFT descriptors to reflect the nuance between the ligand isomers, and even isomer mixtures. Even for the current machine learning

models in organic chemistry, the skeletons chiral ligands are often fixed without directly manipulating the chiral centers (*Science*, 2019, **363**, eaau5631, in which only one chirality of the ligands are considered; *J. Chem. Phys.* **2022**, *156*, 114303). In addition, when considering chirality, the configuration of M(salan) or M(salen) can be complicated: *trans*, *cis- α* and *cis- β* , and the *cis* metal complex can have two chiral configurations (Δ and Λ , see the scheme below). Therefore, we may not use the fragmentation strategy to separate the ligands into two parts. Furthermore, it is also computationally difficult to predict the metal complex configuration even we perform the whole metal complex DFT computation. In brief, from a computation point of view, it remains challenging to incorporate such complicated ligands into the machine learning model at the beginning of the development stage.

(2) Such problem might be resolved to use problem-specific descriptors (*Science*, 2021, **374**, 1134) to cluster chiral bipyrrolidine-based ligands separately from other ligands. Nevertheless, only a small number of bipyrrolidine-based salan ligands (not considering isomer, 5 symmetric ligands and 1 asymmetric ligand, see the scheme below) reported in the literature could be available for the training set. Such number is definitely not enough to perform a cluster-based machine-learning training.

(3) Experimentally, the mechanism of such stereoselectivity control using chiral bipyrrolidine-based salan ligands is enantiomorphic site control (or dual-stereocontrol mechanism reported by Kol and Tolman), which is different from our current chain-end control mechanism. For this type of stereoselective ROP, not only the measurement of P_m/P_r using NMR spectra is necessary, the kinetic studies are often required to determine the chirality preference of the chiral catalyst. Currently, the DFT-descriptor dataset only corresponds to the P_m/P_r value output; whereas for chiral-salan catalyst, we definitely need to incorporate the kinetic selectivity factor s (k_{L-LA}/k_{D-LA}) and even the rates towards *meso*-LA and *rac*-LA into the experimental output sets (polymerization of *meso*-LA by racemic binaphthyl-Al leads to heterotactic PLA instead of syndiotactic PLA, see *J. Am. Chem. Soc.* **2002**, *124*, 1316). Therefore, even from the experiment point of view, the chiral salan ligands should be separated from our current training sets for machine learning as these ligands need additional experiment results for model training and verification.

To this end, given the fact that the current scarce data points for such ligands (as listed in (2)), and the demanding computation work of generating DFT descriptors for different isomeric ligands (as

discussed in (1), we would not include these at the beginning of our work when our primary goal is to establish machine learning workflow for stereoselective polymer catalyst discovery. We and the reviewer are both curious about such stereoselective ROP challenges and fully aware of the importance of such problem, but we must take things methodically rather than being over-ambitious.

All above discussions are added after Supplementary Table 1.

At the end of the manuscript, we also mentioned about the future directions involving developing chiral catalysts, and update the references accordingly: “We expect that in the future, deep learning models (e.g., graph neural networks⁷⁷) could be incorporated into our workflow to enhance search efficiency for more flexible catalyst scaffolds, in the discovery of chiral catalysts for enantioselective polymerization of racemic monomers,⁷⁸⁻⁸² and in the synthesis of stereosequence-defined polymers from mixtures of monomers.⁸³”

Moreover, the authors limit their investigations only to symmetric ligands, thus excluding both asymmetric salen and salan ligands and completely omitting salalen ligands which have proved to be interesting coordinating environments for stereoselective aluminum complexes.

We hope the reviewer understand that our goal in this manuscript is to establish a machine-learning based workflow that can be used to rationally discover high-performance catalysts for stereoselective polymerization and to understand the structural factors that impact catalyst stereoselectivity. There is no prior framework to use the machine learning for the discovery of stereoselective polymerization catalysts, therefore it is unrealistic to start from a library consisting of various asymmetric ligands, which not only increases the difficulty in computation and machine-learning modeling, but also significantly increases synthetic burdens that are opposite to the “accelerated discovery process”, given the fact that our newly discovered symmetric ligands have already worked well just based on the existing literature data as the training set.

For the salalen ligand, here are few reasons that we did not select:

(1) The salalen ligand requests at least three steps to prepare, assuming the salicylaldehyde and one-side protected diamine are commercially available (often not; see *Dalton Trans.* **2013**, **42**, 9279; *J. Am. Chem. Soc.* **2014**, **136**, 2940 also having at least three steps as the bromomethylphenols are not commercially available). This runs counter to our ligand selection rule that the ligand should be synthesized within 3 steps for accelerating the discovery process and for future scale-up.

(2) One challenge for asymmetric ligand is that this increases the computational work for generating DFT descriptors. Using our current fragment method for symmetric salan and salen ligands, we only need to calculate A_m and B_nC_p two parts. While for asymmetric such as salalen, we are expected to calculate A_mB_1 ($n = 1$ for C=N bond), $B_1C_pB_n$, A_mB_n ($n \neq 1$ for *N*-alkyl) three parts (not considering chiral factors). This would dramatically expand the chemical space; however, only less than 20 training data points from the literature could be added into the training library. This will make the model training, optimization and search processes more difficult, as the uncertainty gets increased without a large amount of initial training data to be input into the model.

(3) Studies have shown that salan Al complexes appear to be more active than the salalen Al complexes (*Eur. J. Inorg. Chem.*, **2019**, 2768). This is supported by that many salalen-Al-mediated ROP of *rac*-LA produced polymers with M_{ns} often less than 15 kDa (*J. Am. Chem. Soc.* **2014**, *136*, 2940; *Dalton Trans.* **2011**, *40*, 11469; *Dalton Trans.*, **2013**, *42*, 9279). Notably, ethylene-bridged salalen-Al complexes showed slight isoselectivities, whereas cyclohexane-bridged salalen-Al complexes showed moderate heteroselectivity.

(4) Moreover, some chiral salalen complexes involved somewhat complicated stereoselectivity mechanisms. Kol and coworkers found that the ROP using one salalen-Al complex had a selectivity factor of 10, indicating enantiomeric site control mechanism (*J. Am. Chem. Soc.* **2014**, *136*, 2940). However, the k_{app} for *rac*-LA was similar to that of D-LA and slower than L-LA. The chain-end control seemed to be the prevailing mechanism during enchainment at high monomer conversion, whereas the enantiomeric site control occurred at low monomer conversion. For such complex systems, as we discussed above for the chiral salan ligands, we could not simply incorporate them with other ligands, as more experimental measurements will be needed besides NMR studies. Again, at the beginning stage, we could not introduce such complicated systems with laborious synthetic demands into the machine-learning based workflow.

We also note many other interesting ligand skeletons, such as enolic Schiff-bases ligands, NNO tridentate ligands, catam ligands, etc., were not involved in the initial training dataset. Reasons for exclusion are similar to above discussions on chiral salan ligands, and salalen ligands, especially lack of sufficient literature data points, having too complicated structures to start with, or requiring more kinetic experiments for identification PLA microstructures. We hope that more data-science tools, more problem-specific descriptors, and more efficient algorithms could be added to our system to include all the above chemical knowledge into one holistic model in the near future. All the above discussions are added after Supplementary Table 1.

Reviewers' Comments:

Reviewer #1:

Remarks to the Author:

The authors have made an appropriate revision.
The manuscript is now publishable.

Reviewer #2:

Remarks to the Author:

The authors addressed the comments thoroughly.

Reviewer #3:

Remarks to the Author:

In the revised version of the paper of Huang, Wang et al. some points raised by the reviewers have been clarified/improved. However, according to the reviewer, some points still need modifications/comments.

1) Regarding this comment of Reviewer #3: "the new catalysts proposed by the model are very similar to those reported in the literature and do not bring significant improvements to the already reported results (in terms of stereoselectivity)"

The authors wrote that they disagree with this comment of the reviewer since "our newly discovered catalyst outperformed the literature-reported catalysts". As a fact, the best of their new discovered catalysts shows a Pm of 0.95 (ligand A11C3B1) which is NOT a significant improvement with respect to the Pm of 0.94 reported in the literature (ligand A3C3B1). In addition, the difference between ligands A11C3B1 and A3C3B1 is just a methyl in the R2 position, in place of a tert-butyl, which means that the structures are very similar, as the reviewer wrote.

Regarding the Pr, the authors reported a Pr of 0.89 for the catalyst bearing the A6C1B3 ligand (entry 8 in Table 1) while Gibson reported a Pr of 0.96 (reference 45) for the same catalyst, thus higher than the values found for their new discovered catalysts, which, also in this case, have structures very similar to those already reported.

Moreover, also the choice of limiting the studies to ligands obtainable in 3 synthetic steps is questionable considering that investing more time in the synthesis of a catalyst could be amply repaid by catalysts with better performances in terms of both activity and stereoselectivity.

2) Regarding the comment of Reviewer #3:

"One of the most significant critical points of the model is probably related to the lack of distinction between salen and salan ligands, overlooking the fact that salan ligands introduce new elements of chirality which may be crucial for the course of the reaction."

It is clear that the authors misunderstood the comment, as in the answer they referred to chiral salen and chiral salan ligands, while the reviewer was asking to consider that the binding of achiral salan ligands to aluminum renders the N donors stereogenic (the same does not happen with achiral salen ligands). Different chirality of the nitrogen atoms may also vary the buried volume, and this should be taken into consideration.

Reviewer #1 (Remarks to the Author):

The authors have made an appropriate revision. The manuscript is now publishable.

We thank for the reviewer's kind comments.

Reviewer #2 (Remarks to the Author):

The authors addressed the comments thoroughly.

We thank for the reviewer's kind comments.

Reviewer #3 (Remarks to the Author):

In the revised version of the paper of Huang, Wang at al. some points raised by the reviewers have been clarified/improved. However, according to the reviewer, some points still need modifications/comments.

1) Regarding this comment of Reviewer #3: "the new catalysts proposed by the model are very similar to those reported in the literature and do not bring significant improvements to the already reported results (in terms of stereoselectivity)" The authors wrote that they disagree with this comment of the reviewer since "our newly discovered catalyst outperformed the literature-reported catalysts". As a fact, the best of their new discovered catalysts shows a P_m of 0.95 (ligand A11C3B1) which is NOT a significant improvement with respect to the P_m of 0.94 reported in the literature (ligand A3C3B1). In addition, the difference between ligands A11C3B1 and A3C3B1 is just a methyl in the R2 position, in place of a tert-butyl, which means that the structures are very similar, as the reviewer wrote.

We thank for the reviewer's comments. We would like to emphasize that the purpose of this study is not following the conventional trial-and-error path to look for catalysts with improved stereoselectivity; instead, we developed a machine-learning framework to assist in discovering high-performance catalyst based on the limited literature data points. As we mentioned in our previous comments, the ligand skeleton is fixed, and we only replace substituents on the ligands based on the literature data points. Thus the structural changes can be viewed subtle. Again, we are not focusing on designing brand-new ligands, because we have to establish the machine-learning framework first, before introducing new descriptors or features for innovative ligand skeleton design. The reviewer mentioned the improvement "*in terms of stereoselectivity*". In our previous comments we admitted the P_m value of our is close to the literature data as there is not much room to improve (already have 8 ligands with P_m s over 0.9 and most Al catalysts are isoselective), but only 9 of 56 complexes have P_r s over 0.5. The emphasis of this work is on whether the algorithm can help identify new ligands with high P_m and ligands with high P_r , without going through high-throughput synthesis approach to prepare large amounts of ligands and explore the whole chemical space. We do find a handful of ligands to achieve our goal, especially the heteroselective Al complexes that outperformed the literature data (see below). Thus even the catalyst ligand variation is subtle, the algorithm could still help identify new high-performance catalysts through Bayesian optimization. That actually confirms that our framework is really powerful and has significant potential to recognize subtle structural variations.

Above discussion was added after Supplementary Table 1 accordingly.

Regarding the P_r , the authors reported a P_r of 0.89 for the catalyst bearing the A6C1B3 ligand (entry 8 in Table 1) while Gibson reported a P_r of 0.96 (reference 45) for the same catalyst, thus

higher than the values found for their new discovered catalysts, which, also in this case, have structures very similar to those already reported.

We noted that for Gibson's ($A_6C_1B_3$)Al ligand, the data was obtained at the [LA]/[Al] ratio of 50. We repeated their results at the [LA]/[Al] ratio of 100, as we would like to have relatively high-MW PLA for materials characterization (the oligomers would be difficult to prepare samples for mechanical testing). Our data shows that Gibson's Al complex does have heteroselectivity but the P_r becoming lower when increasing [LA]/[Al] ratio (also the LA conversion rate of 75.8% is moderate) and cannot be used to prepare high-MW poly(*ht*-LA). In our case, our newly identified ligand $A_{16}C_1B_2$ could eventually prepare poly(*ht*-LA) with relatively higher MW (64.7 kDa) and a decent P_r value of 0.87 (Supplementary Table 9 entry 2). As mentioned, the predicted ligand structure is based on the literature ligand data points, they are similar as we did not invent new ligand skeleton or introduce new substituent groups. Even so, our results show the potential that algorithm can assist the rapid discovery of catalysts just based on the known knowledge, and we believe such potential could be beneficial for the whole community to accelerate the discovery without relying on serendipity or experience.

Above discussion on the Gibson's ligand was added after Supplementary Table 4.

Moreover, also the choice of limiting the studies to ligands obtainable in 3 synthetic steps is questionable considering that investing more time in the synthesis of a catalyst could be amply repaid by catalysts with better performances in terms of both activity and stereoselectivity.

We acknowledge that the selection of 3 synthetic-step is arbitrary. Nevertheless, as we have emphasized in our previous comments and the manuscript, time presents a substantial cost in addition to extra reagents. At the beginning of confirming whether our data-science-guided approach is valid, it is burdensome to prepare ligands having 5 synthetic steps without knowing their performance. Besides, our current data have shown that ligands which can be easily made can have excellent performance and have the potential to be scaled up, which clearly justifies our approach.

2) Regarding the comment of Reviewer #3: "One of the most significant critical points of the model is probably related to the lack of distinction between salen and salan ligands, overlooking the fact that salan ligands introduce new elements of chirality which may be crucial for the course of the reaction." It is clear that the authors misunderstood the comment, as in the answer they referred to chiral salen and chiral salen ligands, while the reviewer was asking to consider that the binding of achiral salan ligands to aluminum renders the N donors stereogenic (the same does not happen with achiral salen ligands). Different chirality of the nitrogen atoms may also vary the buried volume, and this should be taken into consideration.

We thank for the reviewer's comments and we acknowledge that the complex between achiral salan and Al can result in stereogenic. The different chiralities / configurations with metal, even for the achiral salan ligand, have been shown in the discussion after Supplementary Table 1 (we already plotted possible 5 configurations). Nevertheless, we did not notice any enantiomorphic-site control mechanism in the polymerization using our new ligands or the literature reported ligands, and our polymerization enchainments are likely mediated by chain-end control mechanism. Note that for enantiomorphic-site control enchainment, the NMR tetrad peaks would be different from chain-end control mechanism, as the stereoerror would be corrected by the chiral catalyst (ideal

enantiomorphic control statistics: $mmr/rmm/rmr/mrm = 1/1/1/2$); and we did not observe such NMR difference.

Above discussion was added after Supplementary Fig. 2

For the buried volume calculation of the ligand, we followed the literature method to calculate the whole catalyst including metal. The Al complex was DFT computed first to obtain the optimized geometry without imaginary frequencies (the lowest energy). As shown in Gibson's paper (Figure 1 in reference 45) and confirmed by our computation studies, there is no isomer found for such achiral salen based Al complexes. Thus, in our case there is only one stable conformation of our Al complex and buried volume calculation is not affected.

As shown in Supplementary Figure 12, we also considered the Al complex configuration when reacting with LA at TS1 state (i.e., changing ligand chirality), and varying the LA docking position (i.e., change the configuration of the Al complex) would lead to increased energy and is not the optimized reaction path. In short, our result is similar to many reports for LA polymerization that even chiral catalyst (or stereogenic catalyst) is used, the enchainment could likely follow chain-end control mechanism.

Above discussion was added accordingly after Supplementary Fig. 12.

Reviewers' Comments:

Reviewer #3:

Remarks to the Author:

The authors addressed all the reviewer's comments.

REVIEWERS' COMMENTS

Reviewer #3 (Remarks to the Author):

The authors addressed all the reviewer's comments.

We thank the reviewer's comments.